# Local vulnerability and global connectivity jointly shape neurodegenerative disease propagation

Ying-Qiu Zheng[1], Yu Zhang[1,2], Yvonne Yau[1], Yashar Zeighami[1], Kevin Larcher[1], Bratislav Misic[1]☯*, Alain Dagher[1]☯*

**1** McConnell Brain Imaging Centre, Montréal Neurological Institute, McGill University, Montréal, Quebec, Canada, **2** Centre de Recherche de l'Institut Universitaire de Gériatrie de Montréal, Montréal, Canada

☯ These authors contributed equally to this work.

* bratislav.misic@mcgill.ca (BM); alain.dagher@mcgill.ca (AD)

**Data Availability Statement:** All public data used here are available to anyone and referenced in the manuscript with the current URLs. The code for running the agent-based model is available at

## Abstract

It is becoming increasingly clear that brain network organization shapes the course and expression of neurodegenerative diseases. Parkinson disease (PD) is marked by progressive spread of atrophy from the midbrain to subcortical structures and, eventually, to the cerebral cortex. Recent discoveries suggest that the neurodegenerative process involves the misfolding and prion-like propagation of endogenous α-synuclein via axonal projections. However, the mechanisms that translate local "synucleinopathy" to large-scale network dysfunction and atrophy remain unknown. Here, we use an agent-based epidemic spreading model to integrate structural connectivity, functional connectivity, and gene expression and to predict sequential volume loss due to neurodegeneration. The dynamic model replicates the spatial and temporal patterning of empirical atrophy in PD and implicates the substantia nigra as the disease epicenter. We reveal a significant role for both connectome topology and geometry in shaping the distribution of atrophy. The model also demonstrates that *SNCA* and *GBA* transcription influence α-synuclein concentration and local regional vulnerability. Functional coactivation further amplifies the course set by connectome architecture and gene expression. Altogether, these results support the theory that the progression of PD is a multifactorial process that depends on both cell-to-cell spreading of misfolded proteins and regional vulnerability.

## Introduction

Neurodegenerative diseases such as Alzheimer disease (AD), Parkinson disease (PD), and amyotrophic lateral sclerosis are a major cause of psychosocial burden and mortality but lack specific therapy. Until recently, the mechanism of progressive neuronal death in these conditions was unknown. However, converging lines of evidence from molecular, animal, and human postmortem studies point to misfolded neurotoxic proteins that propagate through the central nervous system via neuronal connections [1–6]. These pathogenic misfolded disease-specific proteins function as corruptive templates that induce their normal protein counterparts

https://github.com/yingqiuz/SIR_simulator. The deformation maps used to test the model are available at https://neurovault.org/collections/860/.

**Funding:** This research was undertaken thanks in part to funding from the Canada First Research Excellence Fund, awarded to McGill University for the Healthy Brains for Healthy Lives initiative. AD received funding from the Canadian Institutes for Health Research (grant number FDN-143242), Natural Sciences and Engineering Research Council of Canada, Michael J. Fox Foundation, Weston Brain Institute, and the Alzheimer Association. BM acknowledges support from the Natural Sciences and Engineering Research Council of Canada (NSERC Discovery Grant RGPIN 017-04265), the Fonds de recherche Qu ébec–Sant é (Chercheur Boursier), and the Canadian Institutes of Health Research (CIHR; Project Grant 391300). PPMI—a public-private partnership—is funded by the Michael J. Fox Foundation for Parkinson's Research and funding partners, including AbbVie, Avid, Biogen, Bristol-Myers Squibb, Covance, GE Healthcare, Genentech, GlaxoSmithKline, Lilly, Lundbeck, Merck, Meso Scale Discovery, Pfizer, Piramal, Roche, Sanofi Genzyme, Servier, Teva, and UCB. The funders had no role in study design, data collection and analysis, decision to publish, or preparation of the manuscript.

**Competing interests:** The authors have declared that no competing interests exist.

**Abbreviations:** AD, Alzheimer's disease; AHBA, Allen Human Brain Atlas; DBM, Deformation-Based Morphometry; fMRI, functional MRI; GBA, glucocerebrosidase gene; GQI, generalized q-sampling imaging; MNI, Montreal Neurological Institute; PD, Parkinson's disease; PPMI, Parkinson Progression Marker Initiative; QA, quantitative anisotropy; S-I-R, Susceptible-Infected-Removed; SDF, Spin distribution function; SNCA, $\alpha$-synuclein gene.

to adopt a similar conformational alteration, analogous to the self-replication process in prion diseases. Examples include amyloid $\beta$-protein (A$\beta$) and *tau* in AD and $\alpha$-synuclein in PD. The misfolded proteins can deposit into insoluble aggregates and progressively spread to interconnected neuronal populations through synaptic connections. The model of a propagating proteinopathy remains controversial, however [7], and direct evidence in humans remains mostly circumstantial [8].

The prion hypothesis suggests that propagation dynamics in neurodegenerative diseases may be modeled using methods derived from infectious disease epidemiology. Just as infectious diseases spread via social contact networks, misfolded proteins propagate via the brain's connectome. There are different approaches for modeling epidemic spread over a network. In simple diffusion models, disease in any region is modeled as a concentration (e.g., of misfolded protein), and propagation obeys the law of mass effect with first-order kinetics [9, 10].

Such models are easily solved mathematically but have limited explanatory power. Another approach is the agent-based model [11], in which the infectious state of each individual agent and its motility are simulated and in which simple local interactions can translate into complex global behavior. Agent-based models have the advantage of easily incorporating additional emergent properties of a system as the epidemic spreads—for example, a brain region may lose its ability to propagate the disease once it is severely affected. They also easily incorporate differences among agents (e.g., in susceptibility to infection or mobility) and are useful for testing interventions (e.g., vaccination).

Here, we propose a Susceptible-Infected-Removed (S-I-R) agent-based model on a brain network to explore the spreading of pathological proteins in neurodegenerative diseases (Fig 1). The agents are individual proteins. The population is split into "S," the portion yet to be infected (normal proteins), "I," the portion capable of transmitting the infection (misfolded proteins), and "R," the portion no longer active in the spreading (metabolized and cleared proteins). We took PD as an example to show how an S-I-R agent-based model can track the spreading of misfolded $\alpha$-synuclein, the pathological fibrillar species of endogenous $\alpha$-synuclein suggested to be responsible for PD pathology. Although convincing evidence from animal [12–18] and neuroimaging studies [19, 20] supports the propagation of misfolded and neurotoxic $\alpha$-synuclein, other mechanisms may also drive PD pathology, including cell-autonomous factors—dependent on gene expression—that modulate regional neuronal vulnerability [7]. If the pathology of neurodegenerative diseases is indeed driven by progressive accumulation and propagation of disease-related proteins, such a model should recapitulate the spatial pattern of regional neurodegeneration in patients, thereby providing converging and independent evidence for the pathogenic spread hypothesis. We also investigate whether selective vulnerability may influence the spatial patterning of the disease.

## Results

### Model construction

**Structural connectivity.** Diffusion-weighted MRI data from 1,027 healthy participants were used to construct the anatomical network for $\alpha$-synuclein propagation (source: Human Connectome Project, 2017 S1200 release [21]). Adjacency matrices were reconstructed using deterministic streamline tractography [22]. A group consensus structural connectivity matrix was constructed by selecting the most commonly occurring edges averaged across all subjects, resulting in a binary density of 35% [23–25].

**Functional connectivity.** Resting-state functional MRI (fMRI) data from 496 healthy participants (source: Human Connectome Project, 2015 S500 release [21]) were used to construct the functional connectome. Individual functional connectivity matrices were calculated using

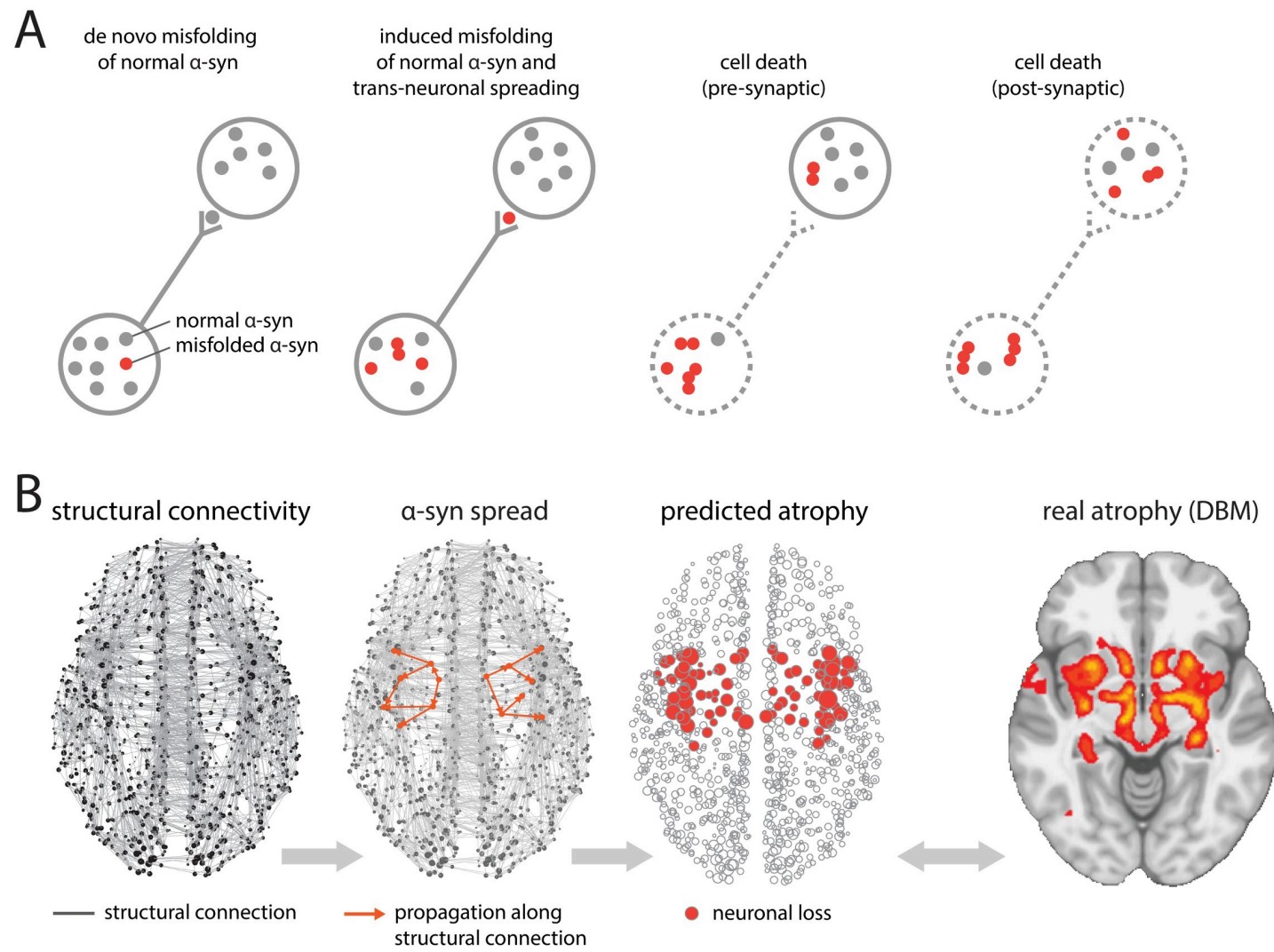

**Fig 1. Agent-based S-I-R model.** (A) Misfolded $\alpha$-synuclein (red) may diffuse through synaptic connections into adjacent neurons, causing misfolding of normal $\alpha$-synuclein (gray). Accumulation of misfolded $\alpha$-synuclein induces neuronal loss. (B) At the macroscopic level, misfolded $\alpha$-synuclein propagates via structural connections, estimated from diffusion-weighted imaging. Simulated neuronal loss (atrophy) is compared against empirical atrophy, estimated from PD patients using DBM. $\alpha$-syn, $\alpha$-synuclein; DBM, Deformation-Based Morphometry; PD, Parkinson disease; S-I-R, Suspectible-Infected-Removed.

Pearson's correlation coefficient and then normalized using Fisher's z transform. A group correlation matrix was then constructed by first averaging the z-score connectivity matrices across subjects, and then converted back to correlation values using the inverse transformation. Negative correlation values in the resultant group connectivity matrix were set to 0.

**Gene expression.** mRNA transcription (measured using in situ hybridization) profiles of the genes $\alpha$-synuclein (*SNCA*) and glucocerebrosidase (*GBA*) were averaged across samples in the same brain parcel and across the 6 subjects in the Allen Human Brain Atlas (AHBA) data set. These gene expression profiles determine the local synthesis and degradation of $\alpha$-synuclein (see Methods).

**Atrophy.** An atrophy map was derived from T1-weighted MRI scans of 237 PD patients and 118 age-matched healthy controls (source: Parkinson Progression Marker Initiative [PPMI] [26]). For each participant (patient or healthy control), the Deformation-Based

Morphometry (DBM) value in each parcel was estimated to quantify the local volume change, on which an unpaired $t$ test was conducted between the patients and healthy controls. The resulting $t$ statistics were taken as the measure of regional atrophy [19].

**Brain parcellation.** The brain MRI template was parcellated according to an anatomical segmentation-based atlas, featuring 68 bilateral cortical and 15 subcortical regions [27–29]. As only 2 of the 6 postmortem AHBA brains have right hemispheric data available, and diffusion tractography is prone to errors in detecting interhemispheric connections, we simulated propagation using only the left hemisphere, yielding 42 regions in total.

**Synuclein propagation.** We posited that regional expression level of endogenous $\alpha$-synuclein already existing in the brain before disease onset may bias the trajectory of misfolded $\alpha$-synuclein propagation. Therefore, to estimate regional density of endogenous $\alpha$-synuclein in the healthy brain, we set up a process that used generic information only to simulate the population growth of normal $\alpha$-synuclein agents. Normal agents in region $i$ are synthesized in each unit area (1 mm$^3$ voxel) per unit time with probability $\alpha_i$ (the synthesis rate in region $i$). Meanwhile, any agent already existing in region $i$ can (a) exit region $i$ and move into the edges it connects to with probabilities proportional to the corresponding connection strengths (densities of the fiber tracts) or (b) remain in region $i$, where it may be metabolized with probability $\beta_i$ (the clearance rate in region $i$). Likewise, the agents in edge $(i,j)$ can (a) exit edge $(i,j)$ to enter region $j$ with probability $1/l_{ij}$ in which $l_{ij}$ is the mean length of the fiber tracts between region $i$ and $j$, reflecting our intuition that agents in longer edges have lower probability of exiting the edge, or (b) remain in edge $(i,j)$ with probability $1 - 1/l_{ij}$. The synthesis rate $\alpha_i$ and clearance rate $\beta_i$ in region $i$ are the *SNCA* and *GBA* expression z-scores, respectively, in region $i$ converted to (0,1) using the standard normal cumulative distribution function. The system has only one stable point that can be found numerically (see S1 Text and S1 Fig), suggesting that the growth of $\alpha$-synuclein will deterministically converge to an equilibrium state set by the connectome and the gene expression profiles. The regional density of normal agents (number of agents per voxel) solved at the stable point was taken as the initial state of the system on which to simulate the misfolded $\alpha$-synuclein spreading process.

**Synuclein misfolding.** We next initiated the pathogenic spread by injecting misfolded $\alpha$-synuclein agents into the seed region, here chosen as the substantia nigra. The updating rules of normal agents (see "Synuclein propagation") were adapted to account for their susceptibility to infection from contact with misfolded agents. Apart from the rules defined in the aforementioned growth process, normal (susceptible) agents in region $i$ that survive degradation can be infected with probability $\gamma_i$, thereby becoming misfolded (infected) agents. In the absence of any molecular evidence to the contrary, misfolded agents are updated with the same mobility (exiting/remaining in regions/edges) and degradation (clearance rate) as normal agents. The new system seeded with misfolded $\alpha$-synuclein has 2 fixed points: (1) one represents the scenario in which misfolded $\alpha$-synuclein dies out, cleared by metabolic mechanisms before being able to transmit the infection to the entire population; (2) the other represents a major outbreak of misfolded $\alpha$-synuclein, spreading to other regions via physical connections, causing further misfolding of endogenous $\alpha$-synuclein and widespread propagation (see S1 Text and S1 Fig). In this model, neither the injection number of misfolded $\alpha$-synuclein agents nor the choice of seed region will affect the magnitude of misfolded $\alpha$-synuclein accumulation at the fixed point; rather, they determine whether the spreading process converges to the epidemic scenario or dies out quickly. See S1 Table for the full list of parameters and their explanations.

## Simulated neuronal loss replicates the spatial pattern of atrophy

We first investigated whether misfolded $\alpha$-synuclein spreading on the healthy connectome could replicate the spatial patterning of atrophy observed in PD patients. We simulated the

propagation of misfolded agents and the accrual of atrophy due to the toxic accumulation of the aggregates. Two factors that may induce neuronal loss were accounted for: (1) the accumulation of misfolded $\alpha$-synuclein, which will cause region-specific cell or synaptic loss directly, and (2) atrophy due to deafferentation secondary to cell death or synaptic loss in connected regions. At each time point, we compared the relative magnitude of simulated atrophy with the spatial pattern of empirical atrophy using Spearman's rank correlation coefficient, yielding the model fit as a function of time $t$.

As the misfolded agents propagate and accumulate in the system, the model fit increases up to a maximum value ($r = 0.63$, $p = 1.71 \times 10^{-5}$, Fig 2A) after which it drops slightly and stabilizes (see S1 Text). It is possible that the slight decrease following the peak occurs because simulated atrophy becomes increasingly widespread as the propagation of misfolded agents progresses, while the empirical atrophy was derived from de novo PD patients at their first visit in PPMI. Fig 2B shows the linear relationship between simulated and empirical atrophy across all nodes at peak fit, while Fig 2C shows the spatial correspondence between simulated and empirical atrophy.

Interestingly, the model fit finally stabilizes with increasing $t$ as the regional accumulation of misfolded $\alpha$-synuclein approximates the stable point (see S2 Fig for model fit up to $10^5$ time steps), a finding that mirrors recent discoveries in animal models in which misfolded $\alpha$-synuclein eventually ceases to increase in later stages [30]. We also note that misfolded $\alpha$-synuclein arrival time at each brain region resembles the well-established Braak stages of PD [31, 32] (S3 Fig). For validation purposes, we estimated the DBM values using an alternative pipeline (*fsl_anat*) [33], reobtained the $t$ statistics as the atrophy measure, and found that the model fit based on the new measure yielded a comparable temporal pattern (S4 Fig).

We next investigated whether the model fit was consistent across variations in structural network connection densities. We selected varying subsets of the most commonly occurring edges in the individual structural connectivity matrices, varying the binary density of the group structural network matrix from 25% to 45% (of all possible edges). We then simulated

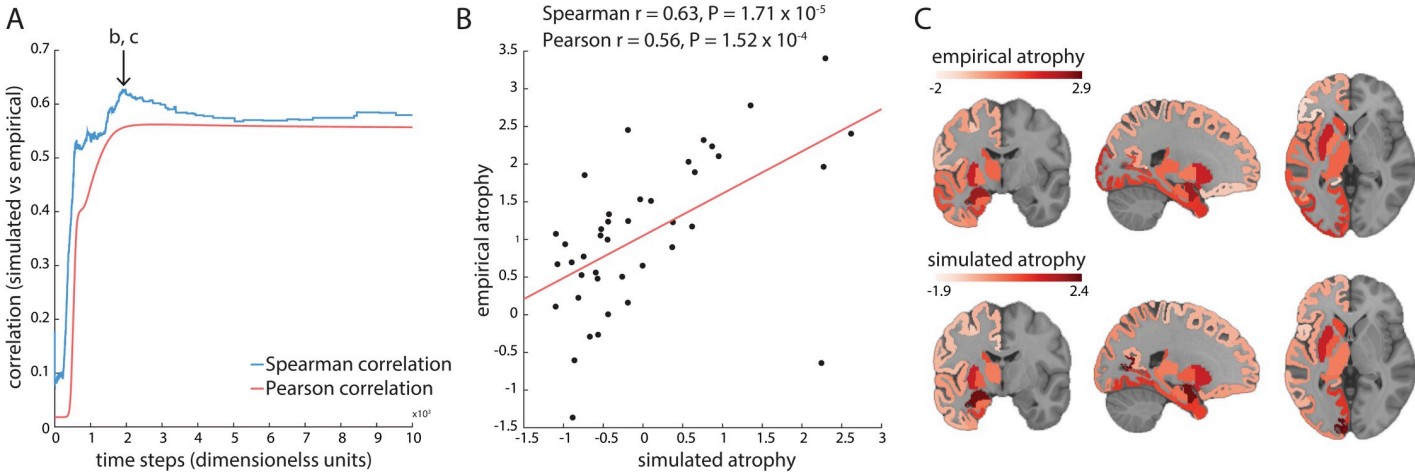

**Fig 2. Model fit.** (A) Correlations between simulated atrophy and empirical atrophy derived from PD patient DBM maps up to $t = 10^4$. Correlations are shown as a function of simulation time. After reaching the peak value ($r = 0.63$, $p = 1.71 \times 10^{-5}$), the model fit slightly drops and finally stabilizes. See S2 Fig for correlations up to $t = 10^5$. (B) Model fit at the peak of Spearman's correlation taken from panel A. Using Pearson's correlation coefficients yielded comparable results ($r = 0.56$, $p = 1.52 \times 10^{-4}$. Values shown in the axes are normalized. The outlier at the bottom right is the nucleus accumbens (for a possible explanation see Discussion). (C) Simulated atrophy and empirical atrophy plotted on the MNI152 standard template. The slices were chosen at x = −22, y = −7, z = 0 (MNI coordinates). The underlying data can be found at https://github.com/yingqiuz/SIR_simulator/blob/master/results/Fig2.mat. DBM, Deformation-Based Morphometry; MNI, Montreal Neurological Institute; PD, Parkinson disease.

the spreading processes on each network, derived the atrophy estimate at each region, and compared it to the empirical atrophy pattern using Spearman's rank correlation coefficient. All the simulations yielded comparable model fits with the peak correlation values consistently around 0.6 (Fig 3, blue curve), suggesting that the S-I-R agent-based model is robust to variations in network density. Notably, we also assessed the Spearman's correlation between the regional density of misfolded $\alpha$-synuclein and the empirical atrophy pattern. Across the same set of networks, simulated atrophy consistently provides better fits with the empirical atrophy than the regional density of misfolded $\alpha$-synuclein (Fig 3, red curve), indicating that accounting for tissue loss due to both $\alpha$-synuclein and deafferentation yields a better model of regional atrophy accrual than the mere accumulation of misfolded $\alpha$-synuclein. Note that, because Spearman's correlation is relatively unstable when sample size is limited, it may peak at early-spreading time frames while, at the time, the simulated atrophy bears no real resemblance to the real atrophy pattern. At the same time, atrophy is a late stage in symptom progression of PD. We therefore discarded the early-spreading time frames, defined as the time steps at which change of misfolded $\alpha$-synuclein density in any of the regions exceeds 1%.

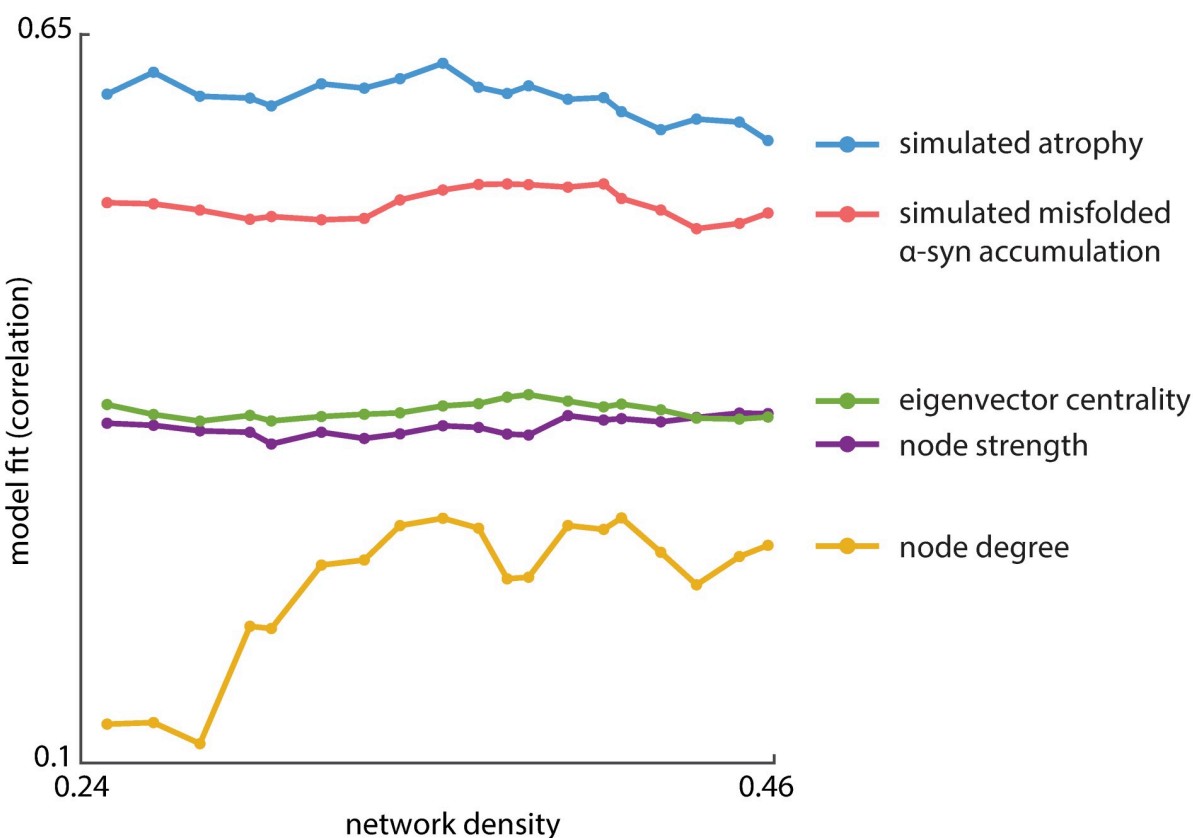

**Fig 3. The full dynamic model outperforms static network measures across multiple network densities.** The full spread model has more predictive power than static topological metrics, including node degree (yellow), node strength (purple), and eigenvector centrality (green). Moreover, simulated atrophy (blue) from the full agent-based model yielded higher correlation with empirical atrophy than the modeled density of misfolded $\alpha$-synuclein (red, peak correlation along $t$ at each density), suggesting that loss of neuronal tissue resulting from misfolded $\alpha$-synuclein accumulation plus deafferentation is a better measure of atrophy in PD than the mere accumulation of misfolded $\alpha$-synuclein. Model fit was assessed using Spearman's correlation coefficient. The overall pattern of results was consistent across multiple network densities. Using Pearson's correlation coefficient yielded similar results (S5 Fig). For the same analysis using two finer-grained anatomical parcellations, see S6 Fig. The underlying data can be found at https://github.com/yingqiuz/SIR_simulator/blob/master/results/Fig3.mat. α-syn, α-synuclein; PD, Parkinson disease.

Finally, we investigated whether the observed atrophy patterns could be directly reproduced from simpler topological measures, without invoking agent-based dynamics. We first tested whether simple regional variation in *GBA* or *SNCA* expression is associated with regional variation in atrophy. Neither the *GBA* nor *SNCA* expression profile bears a strong association with the spatial map of empirical atrophy (*GBA*: Spearman's $r = -0.2402$, $p = 0.1301$; Pearson's $r = -0.3109$, $p = 0.0478$; *SNCA*: Spearman's $r = -0.2385$, $p = 0.1330$; Pearson's $r = -0.2824$, $p = 0.0736$). Next, we tested whether simple network metrics provide a comparable fit to the observed atrophy values. We correlated the atrophy map with node-level network metrics, including node degree, node strength, and eigenvector centrality, at each network density ranging from 25% to 45%. Hubs—or nodes with greater degree connectivity or centrality—tend to be more atrophied (Fig 3, green, purple and yellow curves), echoing the findings that hubs are often implicated in a host of brain disorders [34]. However, none of the metrics performed as well as the full agent-based model in matching the spatial pattern of empirical atrophy. Altogether, these results suggest that the protein dynamics embodied by the S-I-R agent-based model provide explanatory power above and beyond network topology and gene expression.

## Identifying the disease epicenter

We next investigated whether the model yields a disease epicenter consistent with the previous literature. In the aforementioned process of normal $\alpha$-synuclein growth, we solved the regional density of normal agents at the stable point as a baseline estimation of endogenous $\alpha$-synuclein level in healthy brains. Recent findings from animal studies have suggested that $\alpha$-synuclein expression level correlates with neuronal vulnerability in PD [30, 35]; likewise, in our model, higher regional abundance of normal $\alpha$-synuclein agents should indicate greater likelihood of exposure to and growth of infectious agents, higher chance of disease transmission, and, consequently, greater vulnerability to the accumulation of misfolded $\alpha$-synuclein.

We find that, of the 42 left hemisphere regions, substantia nigra has the highest normal $\alpha$-synuclein level (Fig 4, blue line). The elevated density of endogenous $\alpha$-synuclein renders substantia nigra susceptible to the encroaching of infectious misfolded $\alpha$-synuclein in the model, increasing both its vulnerability to misfolded protein and its chance of acting as a disease epicenter to further the propagation of the epidemic. This corresponds with observations of Lewy body inclusions and dopaminergic neuron loss in substantia nigra of PD patients as well as its role in most of the presenting symptoms of the disease [32, 36, 37]. Moreover, other basal ganglia regions have relatively high levels of normal $\alpha$-synuclein at the equilibrium compared to cortical regions (caudate ranks among the highest 42.9% of all the regions; putamen, 31.0%; pallidum, 28.6%), consistent with their role in propagating the disease process to the cerebral cortex [20]. These findings suggest that our model can indeed represent regional variations in selective vulnerability to the pathogenic attacks underlying PD progression by combining information from the healthy connectome and *SNCA* and *GBA* expression.

An alternative definition of disease epicenter is the seed node most likely to propagate an outbreak. As explained in the previous section, the agent-based model has 2 fixed points representing disease extinction or major outbreak. Although in our model the choice of seed region and injection number of misfolded $\alpha$-synuclein agents does not affect the final magnitude of misfolded $\alpha$-synuclein accumulation, it can shift the properties of the two fixed points, determining which one the system will converge to. We posited that the probability of triggering an outbreak in a brain region indicates its likelihood of acting as an epicenter. Therefore, we quantified the spread threshold for each region, i.e., the minimally required injection amount of misfolded $\alpha$-synuclein to initiate an outbreak. In traditional epidemic disease models that do not consider spatial structure or synthesis of new susceptible hosts, basic reproduction

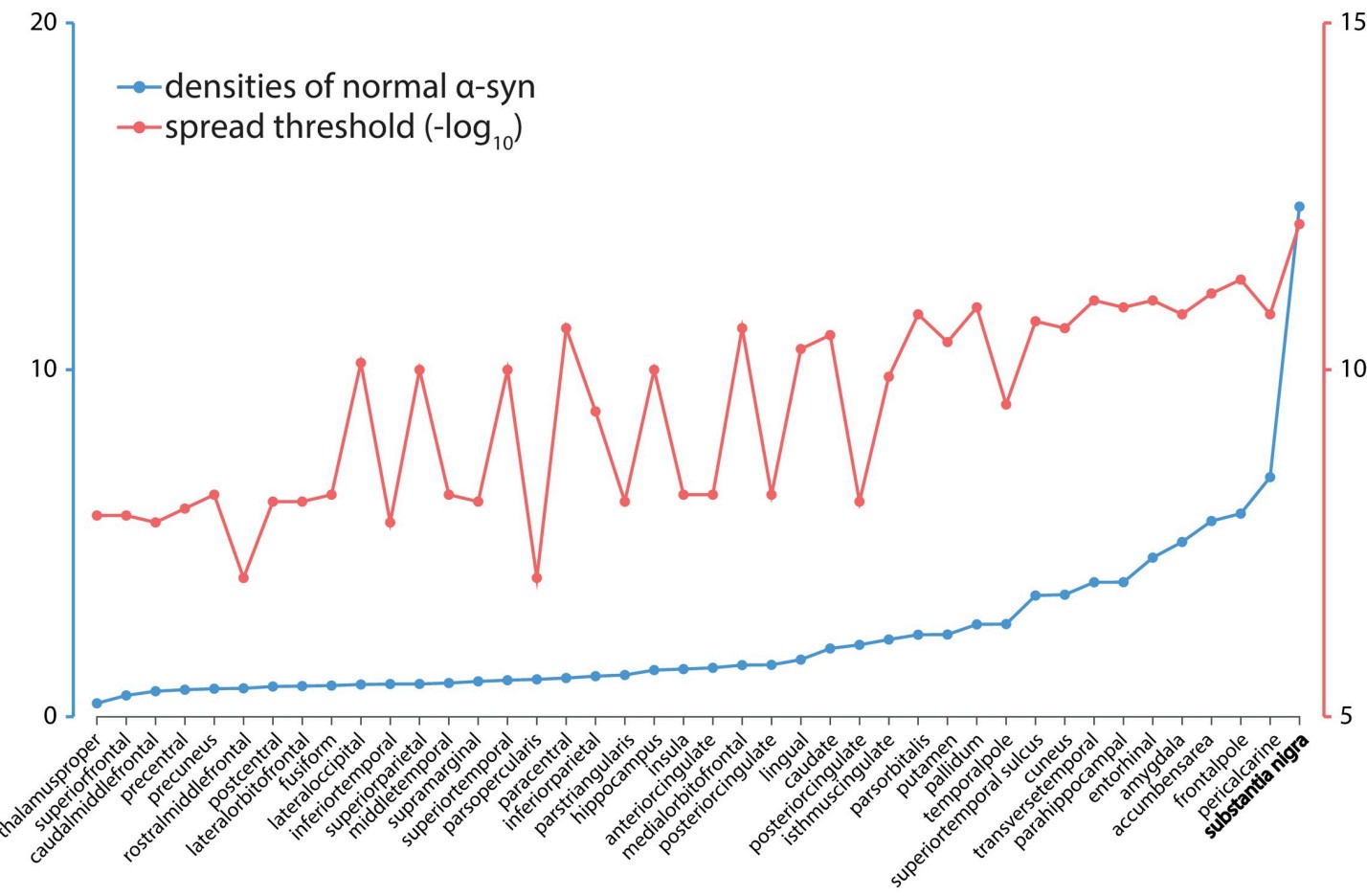

**Fig 4. Identifying the disease epicenter.** Densities of normal $\alpha$-synuclein (blue) at equilibrium (represented by the stable point) and spread threshold (red). Spread threshold was inverted by $-\log_{10}$, so higher values indicate lower thresholds. Spread thresholds reflect the susceptibility of a region to trigger an epidemic. Basal ganglia regions are rich in endogenous $\alpha$-synuclein (caudate ranks among the top 42.9% of regions; putamen, 31.0%; pallidum, 28.6%) and have relatively low spread threshold (caudate ranks among the lowest 35.7%; putamen, 38.1%; pallidum, 16.7%). Substantia nigra has the highest normal $\alpha$-synuclein level and lowest spread threshold, making it the most probable epicenter of disease outbreak. The underlying data can be found at https://github.com/yingqiuz/SIR_simulator/blob/master/results/Fig4.mat. $\alpha$-syn, $\alpha$-synuclein.

number $R_0$ (the average number of susceptible agents that will be affected by an infectious agent before it is removed) marks the transition between the regimes in which disease spreads or extinguishes [38]. However, in our agent-based higher-order system in which new agents are constantly synthesized and move across regions, the transition threshold can only be determined numerically by scanning across different injected amounts of misfolded $\alpha$-synuclein to find the point at which the disease no longer extinguishes. More specifically, starting with an injection amount at which the disease does not spread (here we chose $1 \times 10^{-13}$), we incremented the value by step sizes of $1 \times 10^{-13}$ until the point at which the disease no longer extinguishes, and took this as the spread threshold. This procedure was repeated for every region, yielding 42 regional spread thresholds (Fig 4, red curve).

Substantia nigra has the lowest spread threshold (Fig 4, red curve), suggesting that it is also the most plausible seed region to initiate an epidemic spread. This is consistent with the notion that substantia nigra acts as the epicenter for propagation to the supratentorial central nervous system [19] and is generally one of the earliest regions to display neuronal loss in clinically overt PD. Interestingly, other basal ganglia regions also exhibited relatively low spread

thresholds (caudate ranks among the lowest 35.7% of all the regions; putamen, 38.1%; pallidum, 16.7%). Note, however, that our model does not include regions caudal to the midbrain, which are likely affected earlier than the substantia nigra (see Discussion).

## Connectome architecture shapes disease spread

We next asked whether model fit depends on the connectome's topology and/or spatial embedding (geometry). To address this question, we implemented 2 types of null models, in which (a) the topology of the connectome was randomized (rewired null) or (b) the spatial positions of the regions were shuffled (spatial null) (Fig 5).

Rewired null networks were generated by swapping pairs of edges while preserving the original degree sequence and density using the Maslov-Sneppen algorithm [39] implemented in the Brain Connectivity Toolbox (https://sites.google.com/site/bctnet/) [40]. Note that it is possible that the edges after swapping are not defined in the original connectivity matrices (because no actual fiber tracts exist between the two regions). To interpolate fiber length in the rewired null network, for each region pair $(i,j)$, we calculated the euclidean distances between every possible pair of voxels respectively belonging to region $i$ and $j$ and took the median as the distance between region $i$ and $j$. Next, we fitted a simple linear regression model on the originally existing edges (i.e., $y = w_0 + w_1 x + \varepsilon$, where $y$ is the fiber length and x is the distance (as defined in the section "Synuclein propagation") and assigned the predicted fiber lengths to the new connections created during the rewiring process. Spatial null networks were generated by swapping the physical positions of the nodes while keeping their original connection profiles [41, 42]. This null model retains the degree sequence and connection profiles of every region but randomizes spatial proximity. Networks at binary density 25%, 30%, 35%, and 40% were selected as representatives to construct the 2 types of null networks, with 10,000 realizations each. We then implemented the dynamic model on each network and compared model fits for the empirical and null networks.

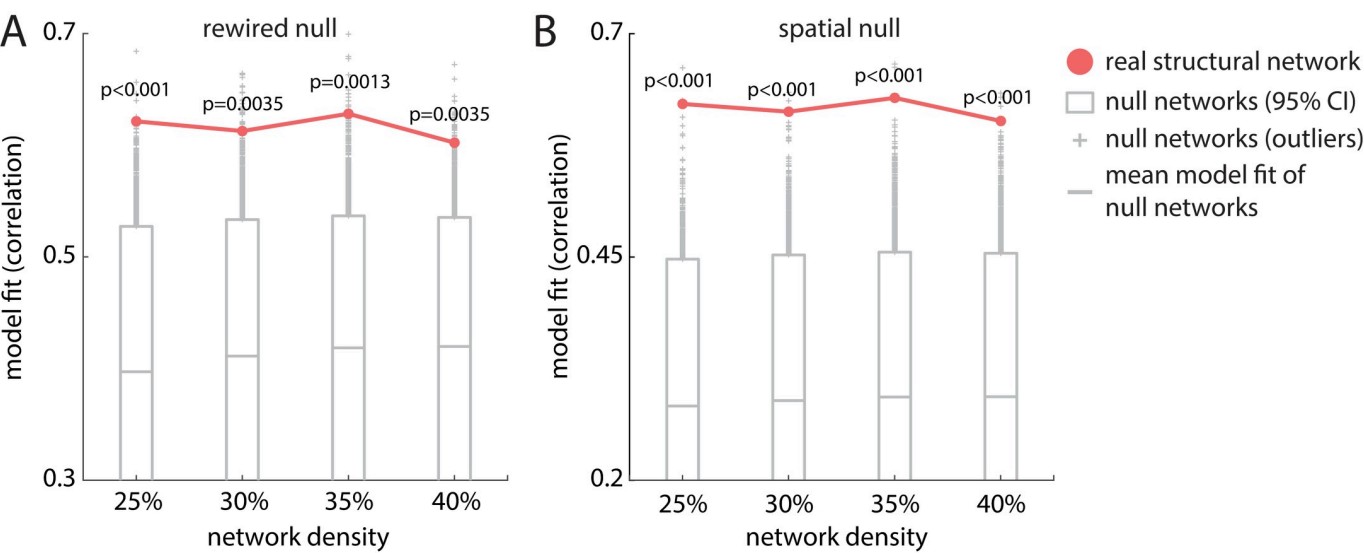

**Fig 5. Effects of network topology and geometry.** (A) Systematic disruption of connectome topology (rewired null). (B) Systematic disruption of spatial embedding (spatial null). Both procedures significantly degrade model fit as measured by Spearman's correlation. Red = real structural network (empirical network); gray = null networks. Rewired null: $p_{25\%} < 0.001$, $p_{30\%} = 0.0035$, $p_{35\%} = 0.0013$, $p_{40\%} = 0.0035$; spatial null: $p_{25\%} < 0.001$, $p_{30\%} < 0.001$, $p_{35\%} < 0.001$, $p_{40\%} < 0.0013$). The underlying data can be found at https://github.com/yingqiuz/SIR_simulator/blob/master/results/Fig5.mat.

The agent-based model simulated on top of the empirical structural network yielded significantly greater fit to empirical atrophy than models simulated on either type of null network. This result was consistent across network densities (rewired null, Fig 5A: $p_{25\%} < 0.001$, $p_{30\%} = 0.0035$, $p_{35\%} = 0.0013$, $p_{40\%} = 0.0035$; spatial null, Fig 5B: $p_{25\%} < 0.001$, $p_{30\%} < 0.001$, $p_{35\%} < 0.001$, $p_{40\%} < 0.0013$) and suggests that the high correspondence between simulated and empirical atrophy in PD is jointly driven by connectome topology and geometry.

### Gene expression shapes disease spread

We next sought to directly assess the influence of local gene expression on spreading patterns of neurodegeneration. Based on molecular evidence, the model uses regional expression of *GBA* and *SNCA* as determinants of $\alpha$-synuclein clearance and synthesis rate. (Note, however, that any other gene known to influence $\alpha$-synuclein synthesis or dynamics could also be included in the model.) Regional *GBA* and *SNCA* expressions were shuffled 10,000 times, respectively, by reassigning the expression scores in each parcel (Fig 6A and 6B, respectively). We then implemented the dynamic models with randomized expression levels and compared differences in model fit when using the empirical gene expression levels (Fig 6, red curve) and permuted gene expression levels (Fig 6, gray bar).

Shuffling the transcription profile of either gene significantly degraded model fit (Fig 6A, *GBA*: $p_{25\%} = 0.0031$, $p_{30\%} < 0.001$, $p_{35\%} < 0.001$, $p_{40\%} < 0.0024$; Fig 6B, *SNCA*: $p_{25\%} = 0.0102$, $p_{30\%} = 0.0201$, $p_{35\%} = 0.0084$, $p_{40\%} = 0.0334$), suggesting a significant role of *GBA* and *SNCA* expression in driving the spatial patterning of atrophy. In other words, the regional expression of the genes, as implemented in the dynamic model, serves to modulate the vulnerability of individual nodes above and beyond their topological attributes by influencing $\alpha$-synuclein synthesis, seeding, and clearance.

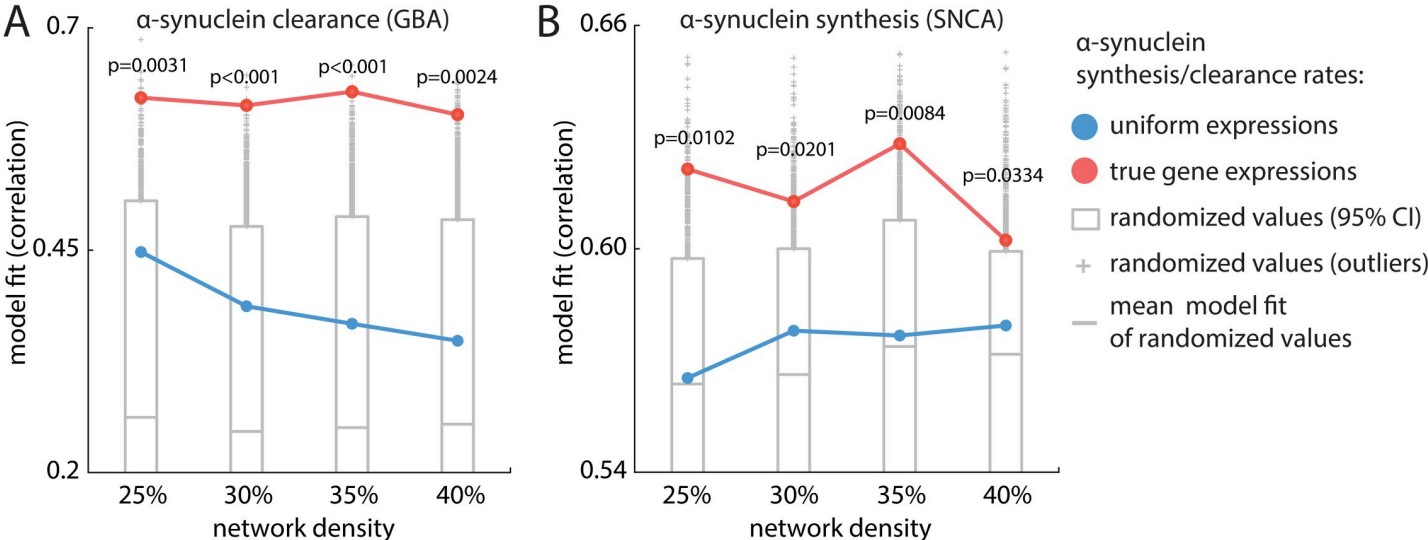

**Fig 6. Assessing the contribution of *GBA* and *SNCA* gene expression.** To assess the influence of gene expression on atrophy, model fit using real expression values (red) is compared to null models in which node-wise expression profiles of *GBA* and *SNCA* (reflecting, respectively, $\alpha$-synuclein clearance and synthesis) were shuffled. Both manipulations significantly reduce model fit regardless of network density. (A) Model fit of randomized *GBA* expression (gray bar) is significantly worse than that of the real *GBA* expression (red line). $p_{25\%} = 0.0031$, $p_{30\%} < 0.001$, $p_{35\%} < 0.001$, $p_{40\%} < 0.0024$. (B) Model fit of randomized *SNCA* expression (gray) is significantly worse than that of the real *SNCA* expression (red). $p_{25\%} = 0.0102$, $p_{30\%} = 0.0201$, $p_{35\%} = 0.0084$, $p_{40\%} = 0.0334$. Notably, uniform transcription profiles, in which all nodes have identical expression values (blue line), yield above-chance model fit but perform poorly compared to the model with real expression values (*GBA* uniform correlations: $r_{25\%} = 0.4479$, $r_{30\%} = 0.3869$, $r_{35\%} = 0.3672$, $r_{40\%} = 0.3481$; *SNCA* uniform correlations: $r_{25\%} = 0.5653$, $r_{30\%} = 0.5780$, $r_{35\%} = 0.5767$, $r_{40\%} = 0.5794$). The underlying data can be found at https://github.com/yingqiuz/SIR_simulator/blob/master/results/Fig6.mat. GBA, glucocerebrosidase gene; SNCA, $\alpha$-synuclein gene.

An alternative explanation for these results is that simply introducing regional heterogeneity in gene expression levels improves model fit, for example, because of differences in general transcription levels between cortex and subcortex. To address this possibility, we further assessed model fit in the cases in which *GBA* and *SNCA* expression is made uniform across all brain regions. Instead of using empirical gene expression, we set uniform synthesis and clearance rates across all regions using the mean expression score, converted to a scalar value between [0,1] using the standard normal cumulative distribution function. We then computed the model fit (peak Spearman's correlation value) for this "uniform" model. The models using uniform transcription profiles underperformed compared to those using empirical transcription profiles (Fig 6, red = empirical gene expression; blue = uniform gene expression); in other words, the incorporation of true local differences in gene expression improves model fit, suggesting that the atrophy pattern in PD is not solely explained by pathogenic spreading per se but also depends on local vulnerability, here dependent on $\alpha$-synuclein concentration. Models implemented using uniform transcription profiles of either gene exhibited above-chance model fit compared to shuffled transcription profiles (*GBA* uniform correlations: $r_{25\%} = 0.4479$, $r_{30\%} = 0.3869$, $r_{35\%} = 0.3672$, $r_{40\%} = 0.3481$; *SNCA* uniform correlations: $r_{25\%} = 0.5653$, $r_{30\%} = 0.5780$, $r_{35\%} = 0.5767$, $r_{40\%} = 0.5794$, blue curve in Fig 6). Altogether, these results demonstrate that regional expression of *GBA* and *SNCA* shapes the spatial patterning of atrophy in addition to connectome topology and spatial embedding.

## Structural and functional connectivity interact to drive disease spread

Finally, we tested whether neuronal activity or pre- and postsynaptic coactivation may facilitate $\alpha$-synuclein propagation. Past neuroimaging studies have shown that cortical thinning in PD is predicted in part by functional connectivity to affected subcortical regions and that regions that exhibit stronger functional connectivity with the substantia nigra tend to exhibit greater atrophy [19, 20]. Secretion of $\alpha$-synuclein by neurons has been shown to be activity dependent [43]. Spread of $\alpha$-synuclein through multiple anatomical pathways may be biased by synchronous activity between the pre- and postsynaptic cells, such that the agents are more likely to move toward regions with higher functional connectivity to a seed region.

To address this question, we integrated resting-state fMRI functional connectivity into the model. We introduce a term $e^{k \times \mathrm{fc}_{(i,j)}}$ to rescale the probability of moving from region $i$ to region $j$ previously defined by the connection strength of edge $(i,j)$ while keeping the sum of the probabilities equal to 1 to preserve the multinomial distribution (Fig 7A). As $k$ is increased, the influence of functional connectivity is greater: stronger coactivation patterns play a more influential role in modulating the motion of the agents on structural connections. For structural edges with relatively small corresponding functional connectivity values, larger $k$ may decrease those edges' contributions to favor propagation through edges with greater functional connectivity. All negative-valued and nonsignificant functional connections were set to 0.

We varied $k$ from 0 (no influence of functional connectivity) to 5 and derived the corresponding peak values of model fit using the same 4 structural connectome densities as before (Fig 7). Model fit was improved by progressively increasing the importance of functional connectivity but only up to a point ($k_{25\%} = 1$, $k_{30\%} = 2.5$, $k_{35\%} = 2.5$, $k_{40\%} = 2.5$). Beyond this point, the influence of functional connectivity dominates the agents' mobility pattern resulting in diminished model fit. The results were consistent across network densities. These results provide evidence for the notion that, while $\alpha$-synuclein propagation and resultant brain atrophy patterns occur via anatomical connections, they may also be biased by neuronal activity.

An alternative explanation is that inclusion of functional connectivity simply leads to overfitting the model. To test this possibility, we investigated whether the same improvement in

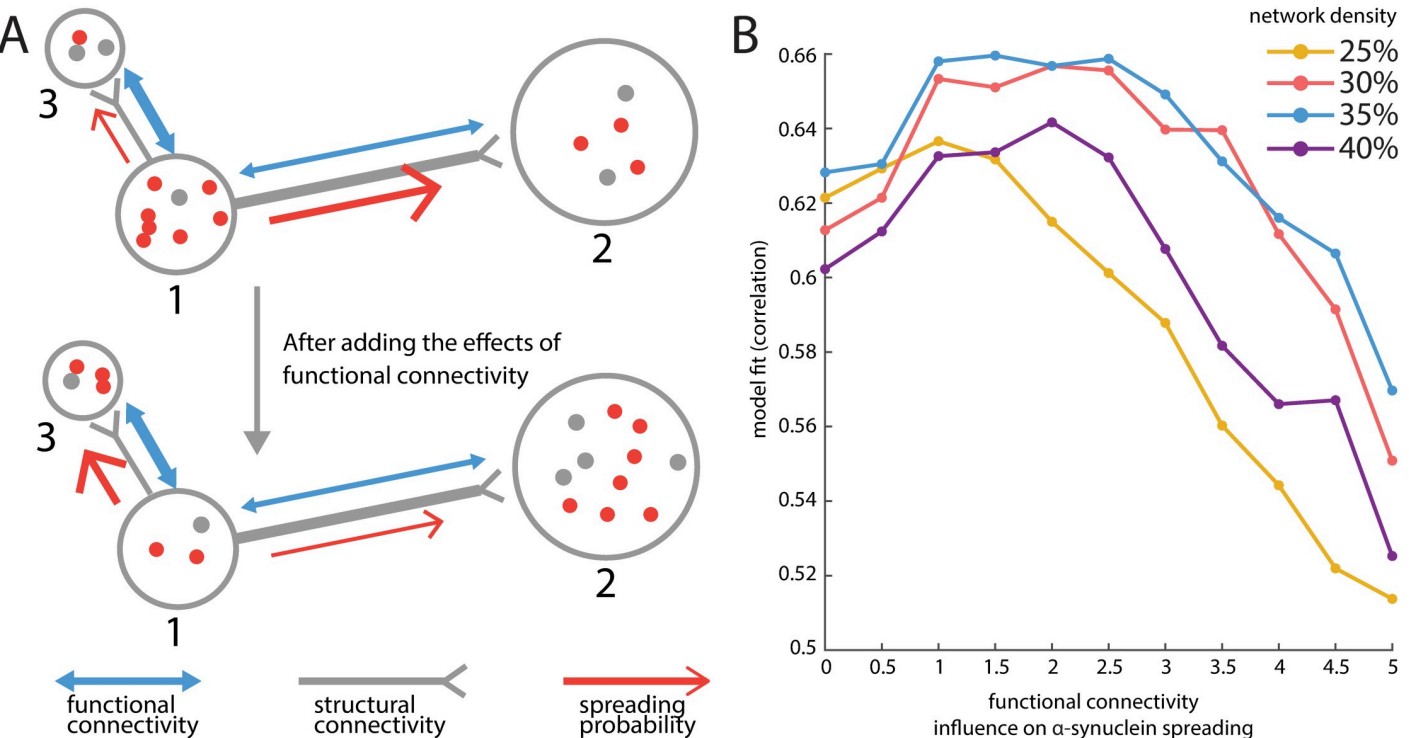

**Fig 7. Incorporating functional connectivity improves model fit.** (A) An illustration of incorporating the influence of functional connectivity. Region 1 is more densely connected with region 2 than with region 3 (i.e., structural connectivity $w_{12} > w_{13}$) but coactivates more with region 3 than with region 2 (i.e., functional connectivity $fc_{12} > fc_{13}$). If functional connectivity is not incorporated, the probability of spreading toward region 2 or 3 for agents in region 1 is proportional to the structural connectivity $w_{12}$ or $w_{13}$ (upper panel); after functional connectivity is incorporated, these probabilities are rescaled to be proportional to $exp\,(k \times fc_{12})w_{12}$ and $exp\,(k \times fc_{13})w_{13}$, respectively (lower panel), in which $k$ is a factor to control the importance of functional connectivity. (B) Resting-state fMRI functional connectivity was incorporated in the model by tuning the probability of $\alpha$-synuclein propagation along structural connections. As the influence of functional connectivity is increased, $\alpha$-synuclein spreading is biased towards structural connections that exhibit high functional connectivity. Model fit is shown for a range of structural connection densities. A balanced effect of functional connectivity and structural connectivity improves model performance, while excessive influence of functional connectivity degrades model fit. The same beneficial effect is not observed when randomized "null" functional connectivity patterns are used (S7 Fig). The underlying data can be found at https://github.com/yingqiuz/SIR_simulator/blob/master/results/Fig7.mat. fMRI, functional MRI.

model fit could be observed if $\alpha$-synuclein spread is biased by randomized functional connectivity patterns. We generated "null" functional connectivity matrices by randomly reassigning the parcellated resting-state fMRI time series into the 42 left hemisphere regions. The results are shown in S7 Fig. We note 2 important results. First, atrophy patterns based on real functional connectivity consistently yield significantly higher model fit than atrophy patterns based on null functional connectivity. Second, model fits based on null functional connectivity do not have the same peaked shape as observed when using real functional connectivity. This further supports the conclusion that atrophy patterns observed in PD patients depend on both the structural and functional architecture of the brain.

## Discussion

### Modeling the spatiotemporal dynamics of neurodegeneration

We developed a networked S-I-R agent-based model of neurodegenerative disease consisting of normal and misfolded proteins. Taking PD as an example, we integrated multimodal neuroimaging and gene expression data to simulate the propagation of misfolded $\alpha$-synuclein on the healthy connectome. The S-I-R agent-based model incorporates pathogenic spread (dominated

by the connectome) and selective vulnerability (modulated here by $\alpha$-synuclein concentration derived from gene expression) under one computational framework. The dynamic model replicated the spatial pattern of measured brain atrophy in PD patients and had greater predictive power than any of the constituent features (i.e., network metrics or gene expression) on their own. Our results demonstrate that connectome topology and geometry, local gene expression, and functional coactivation jointly shape disease progression, as systematic disruption of each of these elements significantly degraded model performance. This model yields insights into the mechanism of PD, providing support for the propagating proteinopathy theory, and can be readily adapted to other neurodegenerative diseases.

The S-I-R model allowed us to map the interaction between network architecture and regional susceptibility and transmissibility. Solving our S-I-R agent-based model numerically yielded 2 fixed (or stable) points of the process after seeding of the infection: rapid extinction or epidemic spread (S1 Text and S1 Fig). If the system is attracted to extinction, misfolded proteins will eventually be cleared. If the system is attracted to the second fixed point, this represents an outbreak. However, in our model, misfolded proteins do not accumulate boundlessly but achieve a stable final concentration at which they coexist with normal proteins. These results are consistent with recent experimental evidence in rodents in which injected misfolded $\alpha$-synuclein grew but eventually ceased to propagate [30], suggesting the existence of an equilibrium. These different outcomes (extinction versus outbreak) might perhaps represent normal aging versus progressive neurodegeneration, or mild versus malignant PD [44].

Agent-based models built on networks allow for the determination of the origin of a disease outbreak and arrival times at different locations [45]. Combining network structure with protein spreading dynamics allowed for the identification of the substantia nigra as the likeliest disease epicenter. Mirroring the reproduction number $R_0$ [38], which marks the transition between disease extinction and outbreak in conventional epidemic models, we estimated spread threshold for our S-I-R agent-based model. This represents the minimum number of infectious agents that need to be introduced in any area to cause an outbreak. In our model, the substantia nigra has the lowest spread threshold, identifying it as a likely disease epicenter. This is not to say that the substantia nigra is the origin of the disease, or the first affected site: the concept of epicenter as used here is similar to "best propagator" [10] and identifies the region most likely to trigger an outbreak rather than the first affected site. According to the Braak hypothesis, the dorsal motor nucleus of the vagus is the initial central nervous system target in PD [31, 32]; however, we could not include structures in the pons and medulla due to difficulty in imaging either atrophy or white matter tracts in the brainstem. Nonetheless, our model is consistent with the substantia nigra acting as a propagator of disease from brainstem to supratentorial areas [19]. We suggest that this may result from its high concentrations of $\alpha$-synuclein and widespread connections. We also used the agent-based model to estimate $\alpha$-synuclein arrival time at each brain region after seeding the substantia nigra (S3 Fig).

We took advantage of several useful features of agent-based models to provide an understanding of factors involved in disease propagation. Others have applied more traditional diffusion models to AD [9, 10] and to neurodegeneration more generally [46]; however the agent-based model used here affords us the possibility of testing different mechanisms of disease, likelihood of outbreak, effect of emergent properties (such as the effect of regional neuronal death on subsequent disease propagation), and eventually, therapeutic interventions. Note that we divided the population into compartments in which the agents share the same characteristics, making the spreading dynamics more tractable and computationally efficient. This simplified model can easily be tailored to accommodate a full agent-based setting by introducing more fine-grained rules. For example, the transmission rate $\gamma_i = 1 - e^{M \, ln(1 - \gamma_i^0)}$ can be

extrapolated as $\gamma_i = 1 - e^{\sum_{k=1}^{M} ln(1 - \gamma_{i,k}^0)}$ to model individually differentiated transmission rates $\gamma_{i,k}^0$ in region $i$.

## Interplay of local vulnerability and network propagation

The small-world properties of brain networks that favor information flow may also potentiate disease spread [47]. These properties include short path lengths [48] and community structure [49, 50], the second of which may potentiate global disease spread by enhancing local, intra-community infection [51]. The presence of high degree nodes (hubs) that are highly interconnected also favors disease propagation [52]. Hubs are expected to have faster arrival times, and greater accumulation of infected agents, making them especially vulnerable to attack. Indeed, hubs manifest greater structural abnormalities in a host of neurodegenerative diseases [34], including PD [19]. Here, we show that disruptions of brain network architecture reduce model fit, providing evidence that the emergent dynamics of synucleinopathy depend on network topology and geometry.

However, while we did find that network metrics predict brain atrophy, the full S-I-R agent-based model provided a better fit to the empirical data than these metrics on their own (Fig 3). Spatial proximity among regions and local differences in synthesis and clearance of $\alpha$-synuclein both impose constraints on the spreading process. As a result, atrophy patterns are shaped by—but ultimately transcend—the underlying connection patterns. The present model correctly predicts that the regions most vulnerable to atrophy are not simply those that participate in the greatest number of connections or those that are a few steps away from other infected regions. More specifically, the agent-based model allowed us to test 2 competing theories of PD pathogenesis: prion-like protein propagation versus regional vulnerability [1, 7]. Here, we chose to model regional vulnerability by incorporating estimated local $\alpha$-synuclein concentration, known to facilitate seeding [53] and increase neuronal vulnerability in animal models [35]. We used regional expression of *GBA* and *SNCA* as estimates of $\alpha$-synuclein clearance and synthesis rates to derive the concentration of endogenous $\alpha$-synuclein. We showed that incorporating this information into the model improved the correlation with empirical atrophy in PD patients; moreover, spatial permutation of gene expression degraded the fit. Thus, our findings support the theory that the dynamics of disease progression arise from an interplay between regional vulnerability and network-wide propagation.

Our results provide converging evidence for the involvement of *GBA* and *SNCA* in PD pathology previously indicated in animal and cellular studies [54]. Mutations in *GBA* are the most common genetic risk factor for PD [55, 56]; mutations and multiplications of *SNCA* have been implicated in driving the severity of the pathology [57–59]. It is worth noting that simple spatial correlation measures alone failed to link *GBA* or *SNCA* regional expression to empirical atrophy; the gene expression effects only emerged from the full agent-based propagating model, which therefore provides a new way to identify gene–disease associations in the central nervous system. New genes can easily be incorporated for the PD model, or to adapt it to other neurodegenerative diseases.

It is also known that $\alpha$-synuclein is secreted in an activity-dependent manner [43]. We therefore tested the influence of resting-state fMRI-derived measures of functional connectivity on protein mobility. As a measure of synchronous neuronal activity in pre- and postsynaptic regions, functional connectivity will bias the proteins into regions showing greater coactivation. Once again, we found that this addition significantly improved the model fit. Thus, functional coactivation also shapes the pattern of disease propagation, explaining why atrophy patterns in neurodegenerative diseases tend to resemble intrinsic functional networks [19, 60].

## Methodological considerations

Although the S-I-R agent-based model provided a good fit to observed neurodegeneration, there are several caveats and limitations in the present study. First, regional variations in vulnerability apart from the effects of $\alpha$-synuclein concentration were not accounted for. It is possible that regions respond differently to the toxicity of $\alpha$-synuclein aggregates, and this can easily be incorporated into the model by introducing new factors, such as genes that control resilience to energetic stress, for example [61]. Moreover, tissue loss was homogeneously modelled as a simple linear combination of local damage (from $\alpha$-synuclein accumulation) and deafferentation, which may not reflect reality. Also, cell death may slow the propagation of misfolded $\alpha$-synuclein and accrual of atrophy, especially in more affected regions. Although we did not take this effect into account here, it can easily be incorporated into the model using agent-based rules.

Note also that our model does not attempt to distinguish between neuronal, axonal, dendritic, or other tissue loss as causes of atrophy. Our only hypothesis is that tissue damage from $\alpha$-synuclein accumulation is reflected in MRI deformation. However, atrophy measured with DBM does not necessarily reflect death of neurons. Indeed, postmortem studies in PD demonstrate a dissociation between $\alpha$-synuclein pathology and neuronal loss, which is prominent in some areas (e.g., substantia nigra) but virtually absent in others (cortex, amygdala) [72]. This is similar to normal senescence, in which there is widespread tissue atrophy in cortex despite preservation of neuronal numbers [73]. Thus, it is possible that tissue atrophy in PD in some regions may reflect loss of dendritic arbors and spines without loss of neurons.

Moreover, the white matter network may not represent the exact physical routes of spread. It is possible that $\alpha$-synuclein spread occurs only between specific cell types—or in one direction—while, in our model, the agents spread bidirectionally along the fiber tracts. The outlier region (accumbens, Fig 2B), which impedes model fit, serves as an example. Nucleus accumbens is one of the least atrophied regions in the data set used here, whereas it exhibits high atrophy in the model. One possible reason for this disagreement is that we did not include the different subsections of the substantia nigra and their projections in the structural connectome used for the model. While we seeded the entire substantia nigra, it is known that the medial portion, which projects to the accumbens [62], is less affected in PD than the lateral substantia nigra, which projects to dorsal striatum [31, 32].

Also, the structural and functional connectomes used here were derived from healthy individuals, as is typically done [9, 10, 19, 60]. However, disease-related alterations to cell integrity should eventually affect network function. Connectomics data in PD are starting to become available, and it will be interesting to incorporate these time-varying effects into our agent-based model.

Finally, we focused on only 2 genes in modelling synucleinopathy, while many other genes such as *LRKK2* and *MAPT* and proteins such as dopamine or *tau* may also influence or interact with synucleinopathy propagation. Using a small subset of genes avoids high model complexity and allowed modelling the proteinopathy in a parameter-free setting. However, the parameter-free setting introduces another caveat: the model converts gene expression scores and fiber density into probabilities without scaling their relative magnitude, while the actual rate of synthesis, clearance, and protein spreading may not be at the same scale. (However, see S8 Fig for evidence that the model is robust to these parameter choices.)

One of the future directions is to customize the model with individual anatomical, functional, genetic, or clinical data to increase its ability to predict disease trajectory and to identify factors that promote resistance to disease spread. Moreover, this model can hopefully help test new preventive procedures. Introducing medications may change the parameters of the

dynamical system; for example, increasing *GBA* activity to elevate the clearance rate would make the stable point for extinction more robust to small perturbations.

# Materials and methods

## Human brain parcellation

We used a brain parcellation generated by atlas-based segmentation [28]. Sixty-eight cortical parcels were defined using curvature-based information [27], which is available at FreeSurfer (http://surfer.nmr.mgh.harvard.edu). Subcortical parcels, including thalamus, caudate, putamen, pallidum, accumbens, amygdala, and hippocamppus, were extracted using the same software from a whole brain segmentation [63]. Finally, substantia nigra was added to the atlas using the location provided in the ATAG atlas (https://www.nitrc.org/projects/atag) [29]. Only the left hemisphere was used in this model, resulting in a total of 42 regions for the subsequent analysis. We used only the left hemisphere to simulate the propagation model because it is difficult to accurately determine interhemispheric connections using tractography [64]. Moreover, regional gene expression was mostly available only for the left hemisphere (see "Regional gene expression").

## PPMI patient data and image processing

PPMI is an open-access comprehensive observational clinical study [26], longitudinally collecting multimodal imaging data, biological samples, and clinical and behavioral assessments in a cohort of PD patients; 3T high-resolution T1-weighted MRI scans of 355 subjects (237 PD patients and 118 age-matched healthy controls) were obtained from the initial visit of PPMI to assess group-level regional atrophy using DBM [19], a method to detect local changes in tissue density. DBM was performed using the minc-toolkit available at https://github.com/BIC-MNI/minc-toolkit-v2.

After denoising [65], inhomogeneity correction [66], and linear intensity scaling, individual MRI images are registered nonlinearly to the MNI152-2009c template [67], yielding the corresponding transformation fields to be inverted into deformation maps in MNI space. Instead of directly using the displacement value $U(x) = (u_1(x), u_2(x), u_3(x))$ of voxel $x$ at coordinates $(x_1, x_2, x_3)$, we calculate the derivative of the displacement in each direction and take the determinant of the jacobian matrix $J$ minus 1, namely, $|J| - 1$, as the value of deformation at $x$, which reflects local volume change.

$$J = \frac{\partial U}{\partial x} = \begin{pmatrix} \frac{\partial u_1}{\partial x_1} & \frac{\partial u_1}{\partial x_2} & \frac{\partial u_1}{\partial x_3} \\ \frac{\partial u_2}{\partial x_1} & \frac{\partial u_2}{\partial x_2} & \frac{\partial u_2}{\partial x_3} \\ \frac{\partial u_3}{\partial x_1} & \frac{\partial u_3}{\partial x_2} & \frac{\partial u_3}{\partial x_3} \end{pmatrix} \tag{1}$$

These values constitute a three-dimensional deformation map for each subject, on which an unpaired *t* test is conducted to derive the statistical difference (*t* score) between the PD patients and the healthy controls at each voxel as a measure of local atrophy. Considering that in the denoising stage a nonlocal smoothing filter was applied to the T1 images, we decided to exclude substantia nigra in the estimation of atrophy because it is too small in size compared to the smoothing parameter and the deformation map may therefore not reflect the true level of tissue loss in such a small structure. Therefore, although substangia nigra is included in the spreading model (and plays a vital role), the model fit was assessed using only the 41 much

larger cortical and subcortical regions (see [19] for more). The deformation maps can be found at https://neurovault.org/collections/860/.

## Regional gene expression

Regional gene expression levels were derived from the 6 postmortem brains included in the AHBA [68], a multimodal atlas of the anatomy and microarray-based gene expression of the human brain. Individuals who donated their brains had no history of psychiatric or neurological disorders. Because 4 of the brains have data from the left hemisphere only, we only modeled the left hemisphere in our study, selecting a total of 3,021 samples of *GBA* (probe ID: 1025373, 1025374) and *SNCA* (probe ID: 1020182, 1010655) in left hemisphere regions. Cortical samples were volumetrically mapped to the 34 cortical regions of our parcellation according to their corrected MNI coordinates (https://github.com/chrisfilo/alleninf) [69], also including samples that are within 1 mm of the nearest gray matter coordinates assigned to any region. Subcortical samples were assigned to one of the 8 subcortical regions as specified by the structure names provided in the AHBA, due to imperfect registration of the postmortem brains onto MNI space. For each probe, all samples that fell in the same anatomical region were averaged and then normalized across all 42 left hemisphere regions, generating transcription maps of each individual probe. These probe maps were next averaged according to the gene classification and normalized again across the regions, yielding the spatial expression profiles for *SNCA* and *GBA*, respectively, represented as $42 \times 1$ vectors (S9 Fig).

## Diffusion-weighted image processing and structural connectivity

A total of 1,027 subjects' preprocessed diffusion MRI data with the corresponding T1 images were obtained from the Human Connectome Project (2017 Q4; 1,200-subject release) to construct an average macroscopic structural connectivity map of the healthy brain. With a multi-shell scheme of b values 1,000, 2,000, and 3,000 s/mm$^2$ and the number of diffusion sampling directions 90, 90, and 90, the diffusion data were reconstructed in individual T1 spaces using generalized q-sampling imaging (GQI) [70] with a diffusion sampling length ratio of 1.0, outputting at each voxel quantitative anisotropy (QA) and the Spin distribution function (SDF), a measurement of the density of diffusing water at different orientations [71].

Deterministic fiber tracking was conducted in native space using DSI studio (https://www.nitrc.org/projects/dsistudio/) [22]. The 42 left hemisphere regions in standard space were mapped nonlinearly onto the individual T1 images using the FNIRT algorithm (https://fsl.fmrib.ox.ac.uk/) [33] with a warp resolution of 8 mm, 8 mm, 8 mm. The 34 cortical regions were dilated toward the gray-white matter interface by 1 mm. The QA threshold was set to 0.6 * Otsu's threshold, which maximizes the variance between background and foreground pixels. To compensate for volume-size introduced biases, deterministic tractography was performed for each region (taken as the seed region) separately. With an angular cutoff of 55, step size of 0.5 mm, minimum length of 20 mm, and maximum length of 400 mm, 100,000 streamlines were reconstructed for each seed region. Connection strength between the seed region and the target region was set to be the density of streamlines (streamline counts) normalized by the volume size (voxel counts) of the target region and the mean length of the streamlines. The goal of this normalization is to correct for the bias toward large regions and long fibers inherent in the fiber tracking algorithms. The procedure was repeated for each region (as the tractography seed region), resulting in 42 connection profiles ($42 \ 1 \times 42$ vectors). Each connection profile consists of the connection strengths between the seed region and all other brain regions with self-connection set to 0. These connection profiles were finally concatenated to generate a $42 \times 42$ structural connectivity matrix per subject. Varying numbers of most commonly

occurring edges were selected and averaged across the individual structural connectivity matrices to construct the group structural connectivity matrix with binary density ranging from 25% to 45%. These group-level matrices were finally symmetrized to represent (undirected) brain networks. Likewise, we also constructed a group-level distance matrix in which elements denote mean euclidean length of the corresponding streamlines, which were used to model the mobility pattern of agents in the edges.

### S-I-R agent-based model

The S-I-R agent-based model includes 5 modules:

a. Production of normal $\alpha$-synuclein

b. Clearance of normal and misfolded $\alpha$-synuclein

c. Misfolding of normal $\alpha$-synuclein (infection transmission)

d. Propagation of normal and misfolded $\alpha$-synuclein

e. Accrual of neuronal tissue loss (atrophy)

It assumes that $\alpha$-synuclein molecules are independent agents with mobility patterns and life spans characterized by the connectome's architecture, neuronal activity, and regional gene expression. The normal $\alpha$-synuclein agents, synthesized continuously under the modulation of regional *SNCA* expression, are susceptible to the misfolding process when they come in contact with a misfolded agent. Once infected, they adopt the misfolded form and join the infected population. Both normal and infected agents may spread via fiber tracts toward connected regions. The degradation rate of both agents is modulated by *GBA* expression, which codes for the lysosomal enzyme glucocerebrosidase [56].

**Production of normal $\alpha$-synuclein.** In each voxel of region $i$, a new normal agent may get synthesized per unit time with probability $\alpha_i$, i.e., the synthesis rate in region $i$. $\alpha_i$ is chosen as $\Phi_{0,1}(SNCA\text{expression}_i)$ where $\Phi_{0,1}(\cdot)$ is the standard normal cumulative distribution function. Therefore, a higher expression score entails a higher $\alpha$-synuclein synthesis rate. The increment of normal agents in region $i$ is $\alpha_i S_i \Delta t$ after a total time $\Delta t$, where $S_i$ is the size (voxel count) of region $i$. $\Delta t$ was set to 0.01.

**Clearance of normal and misfolded $\alpha$-synuclein.** Agents in region $i$, either normal or misfolded, may get cleared per unit time with probability $\beta_i$, the clearance rate in region $i$. As for synthesis rate, $\beta_i$ is set to $\Phi_{0,1}(GBA\text{expression}_i)$. Considering that the probabililty that an agent is still active after a total time $\Delta t$ is given by $\lim_{\delta\tau\to 0}(1 - \beta\delta\tau)^{\Delta t/\delta\tau} = e^{-\beta\Delta t}$, the cleared proportion within time step $\Delta t$ is $1 - e^{-\beta\Delta t}$.

**Misfolding of normal $\alpha$-synuclein (infection transmission).** The normal agents that survive clearance may become infected with probability $\gamma_i = 1 - e^{M_i \, ln(1-\gamma_i^0)}$ in region $i$, where $M_i$ is the population of misfolded agents and $\gamma_i^0$ is the baseline transmission rate that measures the likelihood that a single misfolded agent can transmit the infection to other susceptible agents. Therefore, $(1 - \gamma_i^0)^{M_i}$ is the probability that a single normal agent is not infected by any of the $M_i$ misfoled agents so that $\gamma_i = 1 - (1 - \gamma_i^0)^{M_i} = 1 - e^{M_i \, ln(1-\gamma_i^0)}$ denotes the probability of getting infected by at least one of the $M_i$ misfolded agents in region $i$ per unit time [11, 38]. The baseline transmission rate $\gamma_i^0$ in region $i$ is set to the reciprocal of region size, $1/S_i$. Analogous to the clearance module, the probability that a normal agent is uninfected after a total time $\Delta t$ is given by $\lim_{\delta\tau\to 0}(1 - \gamma_i^0\delta\tau)^{M_i\Delta t/\delta\tau} = e^{-\gamma_i^0 M_i\Delta t}$, therefore the proportion of normal agents that undergo misfolding within $\Delta t$ is $1 - e^{-\gamma_i^0 M_i\Delta t}$.

Therefore, in determining the baseline regional density of normal $\alpha$-synuclein, we increment the population of normal agents $N_i$ with

$$\Delta N_i = \alpha_i S_i \Delta t - (1 - e^{-\beta_i \Delta t}) N_i. \tag{2}$$

After the system reaches the stable point (error tolerance $\varepsilon < 10^{-7}$), we initiate the pathogenic spread and update the population of normal ($N$) and misfolded ($M$) agents with

$$\Delta N_i = \alpha_i S_i \Delta t - (1 - e^{-\beta_i \Delta t}) N_i - (e^{-\beta_i \Delta t})(1 - e^{-\gamma_i^0 M_i \Delta t}) N_i \tag{3}$$

$$\Delta M_i = (e^{-\beta_i \Delta t})(1 - e^{-\gamma_i^0 M_i \Delta t}) N_i - (1 - e^{-\beta_i \Delta t}) M_i. \tag{4}$$

The system has 2 fixed points, the final positions of which will not be affected by the initial conditions of ($N_i$, $M_i$), including the choice of seed region and seeded misfolded agents (see S1 Text). Note that normal and misfolded agents are equivalent to susceptible and infected agents.

**Propagation of normal and misfolded $\alpha$-synuclein.** Agents in region $i$ may remain in region $i$ or enter the edges according to a multinomial distribution per unit time with probabilities

$$P_{\text{region}i \rightarrow \text{region}i} = \rho_i \tag{5}$$

$$P_{\text{region}i \rightarrow \text{edge}(i,j)} = (1 - \rho_i) \frac{w_{ij}}{\sum_j w_{ij}} \tag{6}$$

where $w_{ij}$ is the connection strength of edge ($i,j$) (fiber tracts density between region $i$ and $j$). The probability of remaining in the current region $i$, $\rho_i$, was set to 0.5 for all $i$ (see S9A Fig for other choices of $\rho_i$; we note that the model fit is robust to variations in $\rho_i$). Analogously, the agents in edge ($i,j$) may exit the edge or remain in the same edge per unit time with binary probabilities

$$P_{\text{edge}(i,j) \rightarrow \text{region}j} = \frac{1}{l_{ij}} \tag{7}$$

$$P_{\text{edge}(i,j) \rightarrow \text{edge}(i,j)} = 1 - \frac{1}{l_{ij}} \tag{8}$$

where $l_{ij}$ is the length of edge ($i,j$) (the mean length of the fiber tracts between region $i$ and region $j$). In the absence of definitive molecular evidence of different spreading rates for normal and misfolded $\alpha$-synuclein, we do not assume different exit and propagation dynamics for the two types of agents. We use $N_{(i,j)}$, $M_{(i,j)}$ to denote the normal/misfolded population in edge ($i,j$). After a total time $\Delta t$, the increments of $N_i$, $M_i$ in region $i$ are

$$\Delta N_i = \sum_j \frac{1}{l_{ji}} N_{(j,i)} \Delta t - (1 - \rho_i) N_i \Delta t \tag{9}$$

$$\Delta M_i = \sum_j \frac{1}{l_{ji}} M_{(j,i)} \Delta t - (1 - \rho_i) M_i \Delta t \tag{10}$$

Likewise,

$$\Delta N_{(i,j)} = (1 - \rho_i) \frac{w_{ij}}{\sum_j w_{ij}} N_i \Delta t - \frac{1}{l_{ij}} N_{(i,j)} \Delta t \tag{11}$$

$$\Delta M_{(i,j)} = (1 - \rho_i) \frac{w_{ij}}{\sum_j w_{ij}} M_i \Delta t - \frac{1}{l_{ij}} M_{(i,j)} \Delta t \tag{12}$$

We adopt an asynchronous implementation in which the propagation of normal and misfolded agents is operated before the synthesis, clearance, and infection at each $\Delta t$. We have also tried other implementations, including propagation after synthesis, clearance, or infection at each $\Delta t$ and synchronous implementation and found that the differences are negligible, suggesting that our results are independent of the modules' update order. Note that, although the agent-based model can also be viewed in a stochastic framework (i.e., individual agents alter their states stochastically and the total number of agents at any time is discrete valued), we conducted the simulations in a deterministic way (i.e., using the mean values for each subpopulation of agents in a region, which can take on noninteger values), which preserves the dynamics of disease spreading because the population of protein agents is sufficiently large.

Another important question to consider is whether the model fit that we observed arises only from our particular choice of synthesis, clearance, and propagation rate. Specifying the synthesis and clearance rates as values between 0 and 1 transformed from the gene expression z-scores, we have simplified the complex relationship between transcriptions and the actual function of the gene; likewise, setting the probability of exiting an edge simply to the reciprocal of edge length, we have implicitly specified the relative scale and regional synthesis, clearance, and propagation processes. It is possible that these assumptions implicitly imposed on the model might not be able to reflect the actual spreading process. However, we found that the model yielded robust results as long as the relative magnitude of variations in regional gene expression z-scores is preserved in synthesis and clearance rates (S10A Fig). Moreover, varying the scale of propagation rate with respect to the synthesis or clearance process within a certain range has little effect on model fit as well (S10B Fig).

**Accrual of neuronal tissue loss (atrophy).** We model neuronal tissue loss as the result of 2 processes: direct toxicity from accumulation of native misfolded $\alpha$-synuclein and deafferentation (reduction in neuronal inputs) from neuronal death in neighboring (connected) regions. The atrophy accrual at $t$ within $\Delta t$ in region $i$ is given by the sum of the two processes:

$$\Delta L_i(t) = k_1 (1 - e^{-r_i(t)\Delta t}) + k_2 \sum_j \frac{w_{ji}}{\sum_j w_{ji}} (1 - e^{-r_j(t-1)\Delta t}) \tag{13}$$

where $r_i(t)$ is the proportion of misfolded agents in region $i$ at time $t$, and $1 - e^{-r_i(t)\Delta t}$ quantifies the increment of atrophy caused by accumulation of native misfolded $\alpha$-synuclein aggregates within $\Delta t$ at time $t$. The second term $1 - e^{-r_j(t-1)\Delta t}$, weighted by $w_{ji}/\sum_j w_{ji}$ and summed up across $j$, accounts for the increment of atrophy induced by deafferentation from neighboring regions within $\Delta t$ at t − 1. $k_1$, $k_2$ are the weights of the 2 terms with $k_1 + k_2 = 1$. We set $k_1 = k_2 = 0.5$ such that native $\alpha$-synuclein accumulation and the deafferentation have equal importance in modelling the total atrophy growth (see S10B Fig for other choices of $k_1$, $k_2$; we note that the model fit is consistent across $k_1/k_2$ ranging from 0.1 to 10). Code of the model and relevant data can be found at https://github.com/yingqiuz/SIR_simulator.

## Integration of functional connectivity

We used resting-state fMRI scans from the Human Connectome Project (2015, S500 release) to construct the functional connectivity maps. Both left-right and right-left phase encoding direction data were used. Based on the minimally preprocessed resting-state fMRI data, further processing steps were performed, including (1) nuisance signal regression (including white matter, cerebrospinal fluid, global signal, and 6 motion parameters), (2) bandpass temporal filtering (0.01 Hz f 0.08 Hz), and (3) spatial smoothing using a 4 mm FWHM Gaussian kernel. After quality control, 494 subjects were finally included. We then extracted the mean time course in each of the 42 regions and computed the pairwise Pearson's correlation coefficients to derive individual functional connectivity matrices. Normalized by Fisher's z transform, the functional connectivity matrices were averaged across subjects and converted back to correlations using inverse Fisher transform to generate the group functional connectivity matrix. All negative correlations in the resultant functional connectivity matrix were set to 0, having no influence on the agents' mobility pattern.

Integration of functional connectivity into the model should bias mobility of the agents toward region pairs showing greater coactivation patterns. Agents thus have a higher chance of entering the edges that connect regions having stronger synchronous neuronal activity. More specifically, the weights $w_{ij}$ (connection strength of structural connectivity) in Eq 6 were scaled by $e^{k \times \text{fc}_{(i,j)}}$, where $\text{fc}_{(i,j)}$ is the functional connectivity between region $i$ and region $j$. Therefore, the probability that agents move from region $i$ to edge $(i,j)$ per unit time is determined by

$$P_{\text{region}i \to \text{edge}(i,j)} = (1 - \rho_i) \frac{e^{k \times \text{fc}_{(i,j)}} w_{ij}}{\sum_j e^{k \times \text{fc}_{(i,j)}} w_{ij}} \tag{14}$$

Note that increasing $k$ makes the influence of functional connectivity more differentiated across the edges: the stronger functional connectivity values will be enhanced, while the weaker ones may be suppressed.

## Supporting information

**S1 Text. Analysis of the fixed points.**
(DOCX)

**S1 Table. A list of all the parameters or notations used in the model.** Note that only $k$, $\rho_i$, $k_1$, $k_2$ are free parameters: $k$ was scanned from 0 to 5 to study the effect of functional connectivity on disease spread (Fig 7); $\rho_i = 0.5$ for all the regions so that agents have equal chance of staying in the same region or moving out; $K_1 = K_2 = 0.5$ so that the two factors ([i] native misfolded $\alpha$-synuclein accumulation; [ii] deafferentation from connected regions) contributed equally to the total atrophy growth. We also note that model fit is robust across multiple choices of $\rho_i$, $k_1$, $k_2$ (S10 Fig).
(DOCX)

**S1 Fig. An illustration of the phase plane at α = 5000, β1 = 0.5, β2 = 0.5, γ = 0.001.** M decreases with N (N nullcline, blue, equation [S4]) and N increases with M (M nullcline, orange, equation [S5]), therefore, apart from ($N = 10000$, $M = 0$), there is only one other intersection ($N = 5017.15$, $M = 4982.85$) of the 2 lines, indicating that the system has 2 fixed points only. The vector field (arrows) denotes the direction of the gradient at each position (i.e., the system at that point will move along the direction of the corresponding arrow). The code to generate the figure can be found at https://github.com/yingqiuz/SIR_simulator/tree/master/results/S1_Fig.ipynb.
(TIF)

**S2 Fig. Model fit up to t = 105.** Correlations between simulated atrophy and empirical atrophy derived from PD patient DBM maps. Correlations are shown as a function of simulation time. At large *t*, the model fit stabilizes as the system approaches the stable point. The underlying data can be found at https://github.com/yingqiuz/SIR_simulator/tree/master/results/S2_Fig.mat.
(TIF)

**S3 Fig. Arrival time of misfolded α-synuclein in the model.** (A) Regional arrival time of misfolded α-synuclein is defined as the time steps required for misfolded α-synuclein amount to exceed 1 (after seeding at the substantia nigra with one misfolded agent). This roughly follows the Braak staging hypothesis (see [31]). (B) Arrival time of misfolded α-synuclein at each brain region. The underlying data can be found at https://github.com/yingqiuz/SIR_simulator/tree/master/results/S3_Fig.mat.
(TIF)

**S4 Fig. Model fit with atrophy estimated using fsl_anat.** The computational model replicates empirical atrophy patterns estimated using an alternative DBM pipeline. Model fits are comparable between the *minctools* and *FSL*-estimated atrophy patterns. The underlying data can be found at https://github.com/yingqiuz/SIR_simulator/tree/master/results/S4_Fig.mat.
(TIF)

**S5 Fig. Model fit based on Pearson's correlation coefficient yielded comparable results across network density from 25% to 45%.** The model integrated with gene expression levels has more predictive power than the density of misfolded α-synuclein (red) and the static network metrics, including node degree (yellow), node strength (green), or eigenvector (purple) centrality. The underlying data can be found at https://github.com/yingqiuz/SIR_simulator/tree/master/results/S5_Fig.mat.
(TIF)

**S6 Fig. Model fit at 65-region and 119-region resolution.** The 42 regions used in the main manuscript were hierarchically partitioned into 65 regions and then a further 119 regions. Simulations were conducted on these two finer resolutions, and yielded comparable results to the model fit at 42-region resolution. (A) Spearman's correlation (blue curve) and Pearson's correlation (red curve) versus time using the 65-region parcellation. Black dot: peak position of the correlation coefficients. (B) The model has more predictive power than its constituent factors (as assessed by Spearman's correlation). (C) Spearman's correlation (blue curve) and Pearson's correlation (red curve) versus time using the 119-region parcellation. Black dot: peak position of the correlation coefficients. (D) The model has more predictive power than its constituent factors (as assessed by Spearman's correlation). The underlying data can be found at https://github.com/yingqiuz/SIR_simulator/tree/master/results/S6_Fig.mat.
(TIF)

**S7 Fig. Permutation tests for FC.** Increasing *k* (the influence of FC on α-synuclein transmission) first facilitates then degrades model fit. The red line indicates model fit using true FC values. For each *k*, resting-state fMRI time series were reassigned to construct null FC matrices. The null model fit declines monotonously as *k* increases (gray line). At smaller vaues of *k*, simulations based on real FC yield significantly higher model fit than the null settings as indicated by the 95% confidence interval (gray bar), whereas at larger *k*, real FC ceases to have advantage over null FC. The underlying data can be found at https://github.com/yingqiuz/SIR_simulator/tree/master/results/S7_Fig.mat. FC, functional connectivity.
(TIF)

**S8 Fig. Exploring other choices of synthesis, clearance, and propagation rates.** (A) In the main setting, we transformed the *SNCA* and *GBA* expression z-scores to synthesis and clearance rates using a standard normal cumulative distribution function (Fig 2). To test other possibilities of the relation between gene expression and synthesis or clearance rate, we chose a set of commonly used functions that have domain of all real numbers and return values monotonically from 0 to 1. The temporal pattern of model fit is robust to the choice of transformation functions: the model yields similar results as long as the relative magnitudes of regional gene expressions are preserved. $x_i$ is the gene expression z-score in region $i$ and $f(x_i)$ is the transformation function. (B) We set the probability of exiting an edge $(i,j)$ to the reciprocal of edge length $l_{i,j}$ in the main setting. However, it is possible that the protein agents propagate faster or slower than the regional synthesis or clearance process. To test sensitivity of the model to the rate of protein propagation, we introduced propagation speed $v$ and set the probability thereof to $v/l_{i,j}$ such that varying $v$ changes the relative scale of the propagation process vis-a-vis the regional synthesis and clearance process (e.g., increasing $v$ suggests that the propagation process happens faster than the regional synthesis and clearance processes). We chose $v = 0.1,1,10$ (where $v = 1$ corresponds to the results in the main text) and found that the relative scale of the two processes has little effect on the model fit. https://github.com/yingqiuz/SIR_simulator/tree/master/results/S8_Fig.mat. (TIF)

**S9 Fig. *GBA* and *SNCA* expression.** (A) Regional *GBA* expression. There are 3 probes for *GBA* (probe ID: 1025372, 1025373, and 1025374). Probes 1025372 and 1025373 were included to generate the group transcription profile. Probe 1025374 was excluded as it deviated too much from probe 1025372 (Pearson correlation = 0.30), while the correlation between the other two probes is 0.79. (B) Regional *SNCA* expression. Probes 1020182 and 1010655 were included to generate the group transcription map. Compared to *GBA* expression, *SNCA* is more homogeneous in cortical regions. The underlying data can be found at https://github.com/yingqiuz/SIR_simulator/tree/master/results/S9_Fig.mat. (TIF)

**S10 Fig. Testing free parameters ρi, k1, k2.** Model fit (Spearman's correlation) is robust to variations in ρi, k1, k2 (results shown at network density 35%). (A) $\rho_i$ controls the probability of remaining in region $i$ while $(1-\rho_i)$ is the probability of exiting region $i$ per unit time. The main results are based on $\rho_i = 0.5$. However, the model fit is consistently above 0.55 across $\rho_i$ ranging from 0.1 to 0.9. (B) For the atrophy in region i, $k_1$ controls the contribution of $\alpha$-synuclein accumulation inside region $i$, while $k_2$ controls the contribution of deafferentation induced by atrophy in connected regions. $k_1 + k_2 = 1$. The model fit is consistently over 0.5 across $k_1/k_2$ ranging from 0.1 to 10. These results suggest that the predicative power of the model is robust to variations in free parameters $\rho_i$ or $k_1/k_2$. The underlying data can be found at https://github.com/yingqiuz/SIR_simulator/tree/master/results/S10_Fig.mat. (TIF)

## Acknowledgments

We thank Ross Hammond for fruitful discussions on the idea of applying agent-based models to neurodegeneration.

## Author Contributions

**Conceptualization:** Ying-Qiu Zheng, Yu Zhang, Bratislav Misic, Alain Dagher.

**Data curation:** Ying-Qiu Zheng, Bratislav Misic, Alain Dagher.

**Formal analysis:** Ying-Qiu Zheng, Yashar Zeighami, Bratislav Misic, Alain Dagher.

**Funding acquisition:** Alain Dagher.

**Investigation:** Ying-Qiu Zheng, Bratislav Misic, Alain Dagher.

**Methodology:** Ying-Qiu Zheng, Yu Zhang, Yvonne Yau, Yashar Zeighami, Kevin Larcher, Bratislav Misic, Alain Dagher.

**Project administration:** Alain Dagher.

**Resources:** Alain Dagher.

**Software:** Ying-Qiu Zheng, Yu Zhang, Yashar Zeighami, Kevin Larcher.

**Supervision:** Bratislav Misic, Alain Dagher.

**Validation:** Ying-Qiu Zheng, Bratislav Misic.

**Visualization:** Ying-Qiu Zheng, Bratislav Misic.

**Writing – original draft:** Ying-Qiu Zheng, Bratislav Misic, Alain Dagher.

**Writing – review & editing:** Ying-Qiu Zheng, Yvonne Yau, Bratislav Misic, Alain Dagher.

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
