## [Editor Report · Decision Letter 0]

20 Jul 2019

Dear Dr Dagher, 

Thank you for submitting your manuscript entitled "Connectome architecture, gene expression and functional co-activation shape the propagation of misfolded proteins in neurodegenerative disease" for consideration as a Research Article by PLOS Biology.

Your manuscript has now been evaluated by the PLOS Biology editorial staff, as well as by an academic editor with relevant expertise, and I'm writing to let you know that we would like to send your submission out for external peer review.

**Important**: Please also see below for further information regarding completing the MDAR reporting checklist. The checklist can be accessed here: https://plos.io/MDARChecklist

Please re-submit your manuscript and the checklist, within two working days, i.e. by Jul 23 2019 11:59PM.

Kind regards,

Roli Roberts

Senior Editor

PLOS Biology

INFORMATION REGARDING THE REPORTING CHECKLIST:

PLOS Biology is pleased to support the "minimum reporting standards in the life sciences" initiative (https://osf.io/preprints/metaarxiv/9sm4x/). This effort brings together a number of leading journals and reproducibility experts to develop minimum expectations for reporting information about Materials (including data and code), Design, Analysis and Reporting (MDAR) in published papers. We believe broad alignment on these standards will be to the benefit of authors, reviewers, journals and the wider research community and will help drive better practise in publishing reproducible research. 

We are therefore participating in a community pilot involving a small number of life science journals to test the MDAR checklist. The checklist is intended to help authors, reviewers and editors adopt and implement the minimum reporting framework. 

IMPORTANT: We have chosen your manuscript to participate in this trial. The relevant documents can be located here:

MDAR reporting checklist (to be filled in by you): https://plos.io/MDARChecklist

**We strongly encourage you to complete the MDAR reporting checklist and return it to us with your full submission, as described above. We would also be very grateful if you could complete this author survey:

https://forms.gle/seEgCrDtM6GLKFGQA

Additional background information:

Interpreting the MDAR Framework: https://plos.io/MDARFramework

Please note that your completed checklist and survey will be shared with the minimum reporting standards working group. However, the working group will not be provided with access to the manuscript or any other confidential information including author identities, manuscript titles or abstracts. Feedback from this process will be used to consider next steps, which might include revisions to the content of the checklist. Data and materials from this initial trial will be publicly shared in September 2019. Data will only be provided in aggregate form and will not be parsed by individual article or by journal, so as to respect the confidentiality of responses. 

Please treat the checklist and elaboration as confidential as public release is planned for September 2019.

We would be grateful for any feedback you may have.

---

## [Decision Letter · Decision Letter 1]

6 Sep 2019

Dear Dr Dagher,

Thank you very much for submitting your manuscript "Local vulnerability and global connectivity jointly shape neurodegenerative disease propagation" for consideration as a Research Article by PLOS Biology. As with all papers reviewed by the journal, yours was evaluated by the PLOS Biology editors as well as by an Academic Editor with relevant expertise and in this case by three independent reviewers. The Academic Editor has also added some helpful guidance which can be found at the foot of this email.

Based on the reviews, we will probably accept this manuscript for publication, assuming that you will modify the manuscript to address the remaining concerns raised by the reviewers.

IMPORTANT:

a) Please attend to the concerns raised by the reviewers. However, do note the guidance from the Academic Editor. Essentially her/his suggestion is that Rev #1's request about alternative seeding and Rev #2's requests about dysregulation and PD resting state data should be addressed by discussion alone. HOWEVER, Rev #2's query about sensitivity to parcellation and Rev #3's issue about parameters and model fit must be addressed fully.

b) Please address my Data Policy requests below, namely to supply underlying data and to cite its location in the relevant Figure Legends.

We expect to receive your revised manuscript within two weeks. Your revisions should address the specific points made by each reviewer. In addition to the remaining revisions and before we will be able to formally accept your manuscript and consider it "in press", we also need to ensure that your article conforms to our guidelines. A member of our team will be in touch shortly with a set of requests. As we can't proceed until these requirements are met, your swift response will help prevent delays to publication.

Please note that you may have the opportunity to make the peer review history publicly available. The record will include editor decision letters (with reviews) and your responses to reviewer comments. If eligible, we will contact you to opt in or out.

Early Version: Please note that an uncorrected proof of your manuscript will be published online ahead of the final version, unless you opted out when submitting your manuscript. If, for any reason, you do not want an earlier version of your manuscript published online, uncheck the box. Should you, your institution's press office or the journal office choose to press release your paper, you will automatically be opted out of early publication. We ask that you notify us as soon as possible if you or your institution is planning to press release the article.

Sincerely,

Roli Roberts

Senior Editor

PLOS Biology

ETHICS STATEMENT:

The Ethics Statements in the submission form and Methods section of your manuscript should match verbatim. Please ensure that any changes are made to both versions.

-- Please include the full name of the IACUC/ethics committee that reviewed and approved the animal care and use protocol/permit/project license. Please also include an approval number if one was obtained.

-- Please include the specific national or international regulations/guidelines to which your animal care and use protocol adhered. Please note that institutional or accreditation organization guidelines (such as AAALAC) do not meet this requirement.

-- Please include information about the form of consent (written/oral) given for research involving human participants. All research involving human participants must have been approved by the authors' Institutional Review Board (IRB) or an equivalent committee, and all clinical investigation must have been conducted according to the principles expressed in the Declaration of Helsinki.

DATA POLICY:

We note that the model code is in Github and the deformation maps in Neurovault; we also assume that some of the figure panels are generated directly from model outputs or raw-ish data. However, we also require the simpler structured data that is presented in graphs to be made available in one of the following forms:

Regardless of the method selected, please ensure that you provide the individual numerical values that underlie the summary data displayed in Figs 2B, 3, 4, 5, 6, 7, S3C, S4, S5, S6 and S9, as they are needed for readers to assess your analysis and to reproduce it. Please also ensure that figure legends in your manuscript include information on where the underlying data can be found.

REVIEWERS' COMMENTS:

Reviewer #1:

Thank you for inviting me to review this manuscript by Zheng and colleagues, in which the authors use an inventive combination of agent-based modeling and multimodal neuroimaging approaches to examine the trans-synaptic spread hypothesis in individuals with Parkinson's disease. The conception and execution of the analyses in the study are exemplary -- indeed, I have no issues to raise with the methodological aspects of the study. My only (admittedly minor) concerns lie with the interpretation of the results. In brief, I consider the lack of detailed brainstem anatomy as a crucial missing piece of the story, and worry that the authors may be slightly over-interpreting their results without due appreciation of this issue. Importantly, I don't think that this fact should impede publication of this important work, but I do maintain that the work would be more useful to the broader community if the issues associated with the missing/hidden nodes in the brainstem was given more consideration in the manuscript.

I have included below comments which I hope are helpful.

* The authors chose to 'seed' the substantia nigra, as it is commonly identified as a crucial nexus of pathology in Parkinson's disease. While I don't doubt that this is the case, there is a long-standing and influential hypothesis (Braak, 200) that proposes that alpha-synuclein pathology begins in either olfactory cortex or the ventral medulla (likely through vagal afferents from the peripheral nervous system). Would the authors expect to see similar results if they seeded a different region that was more consistent with the known pathological process at play in PD? Or does the inherent uncertainty associated with creating accurate white-matter connectomes in the brainstem limit this line of questioning? Either way, I think that it's important that this issue is argued/justified in the manuscript, lest the reader come away with the opinion that alpha-synuclein pathology somehow manifests in the substantia nigra without affecting other regions/nuclei.

* I worry that few (if any) brainstem structures other than the SN were included in the authors model. Based on the patterns of known pathology, I would expect that connectivity within and between different subregions of the brainstem would have a major influence over patterns of pathological spread within the cortical and subcortical network of the brain.

* Along similar lines, I worry that the focus on the substantia nigra unnecessarily limits the scope of the authors conclusions. What would happen if you seeded another region that is associated with substantial lewy pathology in PD, such as the locus coeruleus, dorsal raphe, pedunculopontine nucleus, to name but a few? Could seeding these regions better explain spread/atrophy associated with some of the other symptoms of the Parkinson's syndrome, such as cognitive impairment, mood disturbance and visual hallucinations?

* On page 10, the authors claim that their results are consistent with the Braak hypothesis, and that the SN is one of the earliest sites affected. I'm not sure that this is entirely accurate -- see Figure 1A of Surmeier, Obeso and Halliday, 2017 (https://www.nature.com/articles/nrn.2016.178), which shows that SNc is only affected in Braak Stage III/IV. By that stage, many other regions in the brainstem, forebrain and subcortex are affected, suggesting that the SN is not necessarily that important for disease spread per se (though it may be [and likely is] for motor symptoms].

Reviewer #2:

In this manuscript, Zheng and colleagues report on an agent-based model, incorporating multiple aspects PD pathophysiology, aiming for predicting the brain regions first affected by neuronal loss. The approach is highly innovative and novel, as for the first time, multiple divergent parameters are taken into account, which were previously not incorporated into one model. The authors integrate structural connectivity signatures, gene expression data as well as resting state functional connectivity. The model seems to be able to replicate the typical course of PD disease development in terms of regions first affected by neuronal loss, and, identifies the substantia nigra as regions with highest vulnerability. While I cannot judge the merits of the model in mathematical terms, it seems to overcome limitations of previous simpler models, such as diffusion models, and seems to be ideally suited to simulate and test the relatively novel hypothesis of a seed-based disease course, spreading throughout the network. Notably, they attribute at least theoretical evidence for the hub-neuron hypothesis. The manuscript is well written, and of high interest particularly outside of the computation neuroscience community. However, it falls short on three essential aspects:

1. The authors focus on the endpoint of local and global disease development: neuronal loss. All of the benchmarks the authors present, are late-stage human PD data, in which regional atrophy is apparent. While this might still be meaningful, it does not at all reflect the current view of disease development. We know now, that much prior to neuronal loss, the network becomes dysregulated, and these dysregulations often entail local hyperactivity. This hyperactivity itself then might lead to subsequent neuronal loss, and reflect back on the molecular disease entity, please see e.g. Iaccarino et al,, Nature 2016, and Zott et al, Ann rev Neur 2018. These dysregulated network components are not necessarily identical to those networks which degenerate first. The authors need to extend their approach to incorporate these current developments. Ultimately, it is of interest to identify the dysregulated hubs, prior to irrevocable neurodegeneration.

2. Related to my first comment, the authors use a resting state data from healthy controls. How can the authors relate their model to a functional connectome, which does not reflect the connectome of a PD patients? They should incorporate the resting state signatures of different stages of PD disease development.

3. They authors used a parcellation approach, which certainly makes sense. However, could the authors comment on the stability of their model, if the number of parcellations is increased or decreased?

Reviewer #3:

Summary:

Authors develop an agent-based epidemic (SIR) model to represent the spread of pathological proteins in neurodegenerative diseases.

Their approach integrates structural connectivity, functional connectivity and gene expression, to predict sequential volume loss due to neurodegeneration. 

1. Their model concludes implicates the substantia nigra as the “disease” epicenter.

2. Their model concludes that SNCA and GBA transcription influence alpha-synuclein concentration and local regional vulnerability.

3. Their model concludes that functional co-activation further amplifies the course set by connectome architecture and gene expression.

Comments:

1. The choice of an epidemic model seems well-motivated in the Intro.

2. It is evident that this work represents a thorough and thoughtful effort in bringing together several important datasets. All datasets and code have been made available to the community, and the authors' insights are a nice jumping-off point for continued exploration (by them, or others).

Suggestions:

Ultimately, the model fit is not overly convincing, in particular given the number of parameters tuned - peaking at correlation 0.6. That said, I personally do not feel that this fact detracts from the importance or publish-ability of this work. Authors have made explicit how they selected their parameters, and follow-up work can explore variations thereof. However, I think authors could include a brief discussion of the "goodness" of this result, in their view, given broader context.

COMMENTS FROM THE ACADEMIC EDITOR:

My take is that both R1 and R2 are essentially requiring relatively minor revisions.

I think that it is clear that R1 demands that the authors requires alternative seeding (and therefore brain stem anatomy) should be taken on board as a discussion issue. This discussion will lay the basis of a future study.

My interpretation of R2 is much along the same lines. In R2's first point, addressing dysregulation (rather than neuron loss) would be much closer to addressing the onset of the disease and therefore the factors that are more directly involved in the pathological spreading (which is the long-term aim of the study).

Likewise R2's second point is coming back to disease onset by asking that the authors should use resting state data from PD patients (preferably in early onset of the disease). These are two very valid points that I think will constitute a future refinement of the model that will find itself in a future study.

HOWEVER, varying the parcellation (R2 point 3) directly applies to the present study and needs to be addressed. As does R3's issue of choice of parameter and goodness of fit of the model.

In summary I feel that should the paper be revised by including the additional analysis suggested in R1 and R2 it will be come overtly cumbersome and counter productive. However, a full discussion of the issues raised by R1 and R2 will greatly help elucidate the perspectives of the study.

---

## [Editor Report · Decision Letter 2]

31 Oct 2019

Dear Dr Dagher,

On behalf of my colleagues and the Academic Editor, Henry Kennedy, I am pleased to inform you that we will be delighted to publish your Research Article in PLOS Biology. 

Early Version

PRESS 

Kind regards,

Sofia Vickers

Senior Publications Assistant

PLOS Biology

On behalf of, 

Roland Roberts,

Senior Editor

PLOS Biology